

# Climate model configurations of the ECMWF Integrated Forecast System (ECMWF-IFS cycle 43r1) for HighResMIP

Christopher D. Roberts[1,2], Retish Senan[1], Franco Molteni[1], Souhail Boussetta[1], Michael Mayer[1], and Sarah Keeley[1]

[1]European Centre for Medium-Range Weather Forecasts, Shinfield Park, Reading, RG2 9AX, United Kingdom
[2]Met Office, Fitzroy Road, Exeter, EX1 3PB, United Kingdom

*Correspondence to:* C.D. Roberts (chris.roberts@ecmwf.int)

**Abstract.** This paper presents atmosphere-only and coupled climate model configurations of the European Centre for Medium-Range Weather Forecasts Integrated Forecast System (ECMWF-IFS) for different combinations of ocean and atmosphere resolution. These configurations are used to perform multi-decadal ensemble experiments following the protocols of the High Resolution Model Intercomparison Project (HighResMip) and phase 6 of the Coupled Model Intercomparison Project (CMIP6). These experiments are used to evaluate the sensitivity of major biases in the atmosphere, ocean, and cryosphere to changes in atmosphere and ocean resolution. Climatological surface biases in ECMWF-IFS are relatively insensitive to an increase in atmospheric resolution from ∼50 km to ∼25 km. However, increasing the horizontal resolution of the atmosphere while maintaining the same vertical resolution enhances the magnitude of a cold bias in the lower stratosphere. In coupled configurations, there is a strong sensitivity to an increase in ocean model resolution from 1° to 0.25°. However, this sensitivity to ocean resolution takes many years to fully manifest and is not apparent in the first year of integration. This result has implications for the ECMWF coupled model development strategy that typically relies on the analysis of biases in short (< 1 year) ensemble (re)forecast data sets. The impacts of increased ocean resolution are particularly evident in the North Atlantic and Arctic, where they are associated with an improved Atlantic meridional overturning circulation, increased meridional ocean heat transports, and more realistic sea-ice cover. In the tropical Pacific, increased ocean resolution is associated with improvements to the magnitude and asymmetry of ENSO variability and better representation of non-linear SST-radiation feedbacks during warm events. However, increased ocean model resolution also increases the magnitude of a warm bias in the Southern Ocean. Finally, there is tentative evidence that both ocean coupling and increased atmospheric resolution can improve teleconnections between tropical Pacific rainfall and geopotential height anomalies in the North Atlantic.

## 1 Introduction

The European Centre for Medium-Range Weather Forecasts (ECMWF) uses a global general circulation model known as the Integrated Forecast System (IFS) to produce probabilistic ensemble forecasts at lead times of several days to one year ahead.



Since the implementation of cycle 43r1 in November 2016, the operational version of the IFS used to make medium-range ensemble forecasts has included dynamic representations of the atmosphere, ocean, sea-ice, and ocean waves (Buizza et al., 2017).

Scientific development of the IFS is focussed on the improved representation of processes that are important for predictability on operational time-scales. For this reason, emphasis is placed on the identification of systematic biases associated with 'fast' processes that are important for numerical weather prediction (Rodwell and Palmer, 2007). Such errors are typically studied by examining biases in (re-)forecasts at different operational lead times, scrutiny of the analysis increments available from data assimilation systems, and evaluation of forecast reliability (Rodwell and Palmer, 2007; Palmer et al., 2008).

In order to identify errors associated with 'slow' processes it is necessary to run longer climate integrations. Such experiments can provide complementary information to that available from short-range forecasts and may provide additional constraints on the representation of physical processes that are important in forecast mode. For example, multi-decadal experiments enable the investigation of asymmetries and time-scales of coupled ocean-atmosphere phenomena such as the El Niño-Southern Oscillation (ENSO) (see section 4.1), which provides context for the interpretation of biases in seasonal forecasts. In addition, free-running climate integrations are an important tool for understanding the behaviour and deficiencies of multi-decadal reanalyses, particularly when they cover periods and/or regions with limited observational constraints (e.g. Hersbach et al., 2015; Poli et al., 2016).

The longest experiments performed routinely at ECMWF are 13 month integrations to assess the climatology of each new IFS cycle. However, there are efforts in the wider scientific community to evaluate the behaviour of the ECMWF model on longer time-scales. In particular, the EC-Earth consortium (Hazeleger et al., 2010) develop and maintain a global climate model configuration that shares many components with the IFS. However, the latest version of the EC-Earth model has been developed starting from a previous IFS cycle that was operational several years ago, due to the different time-scales involved in the preparation of new operational IFS cycles (∼6 months) and new EC-Earth configurations intended for climate applications (∼5 years). The operational and EC-Earth versions of the IFS also differ in their coupling strategy, tuning of model components, ocean and sea-ice model versions, and the treatment of ocean waves. Because of these distinctions, it is not always straightforward for lessons learned in EC-Earth experiments to be directly applied to the most recent operational IFS cycles.

In this paper, we describe climate model configurations of the IFS developed under the auspices of the European Research Council Horizon 2020 PRIMAVERA project (PRIMAVERA website, 2017) that are built upon IFS cycle 43r1 and follow the protocols defined by the High Resolution Model Intercomparison Project (HighResMip; Haarsma et al., 2016) and phase 6 of the Coupled Model Intercomparison Project (CMIP6; Eyring et al., 2016). In addition, we present results from multi-decadal coupled and atmosphere-only experiments and provide an initial assessment of the the impact of resolution and coupling in climate integrations that are traceable to a recent version of the ECMWF weather forecast model.

The rest of this manuscript is organized as follows: Section 2 describes the model configurations with a focus on the differences compared to the standard operational configuration of IFS cycle 43r1. Section 3 presents an initial assessment the major biases in the atmosphere, ocean, and cryosphere and includes some comment on the impacts of coupling and resolution.





Section 4 is focussed on the representation of ENSO variability and its associated asymmetries, non-linear ocean-atmosphere feedbacks, and global teleconnections. Discussion and conclusions are presented in section 5.

## 2  Model configuration

The physical sub-models of the ECMWF forecast model (atmosphere, sea-ice, ocean, land surface, ocean-waves) and the in-
frastructure required to perform data assimilation and ensemble forecasts are collectively known as the Integrated Forecast System (IFS). Full documentation for the operational version of IFS cycle 43r1 is available online (ECMWF website, 2017). This section provides a brief summary of the main model components and details any differences relative to the operational configuration of IFS cycle 43r1. Following the nomenclature adopted within the PRIMAVERA project, we refer to the configuration presented here as ECMWF-IFS. Low- and high-resolution configurations are referred to as ECMWF-IFS-LR and
ECMWF-IFS-HR, respectively and experiments forced with observed sea surface temperature (SST) and sea-ice concentrations are identified with the suffix 'A' (e.g. ECMWF-IFS-LRA). We also present selected results from a 'mixed resolution' configuration that we term ECMWF-IFS-MR, which consists of the atmospheric model from ECMWF-IFS-LR coupled to the ocean model from ECMWF-IFS-HR.

### 2.1  Atmosphere model

The atmospheric component of the IFS is based on a hydrostatic, semi-Lagrangian, semi-implicit dynamical core with computations alternated between between spectral and reduced Gaussian grid-point representations each time step (Hortal and Simmons, 1991; Hortal, 2002; Ritchie et al., 1995; Simmons et al., 1989; Temperton, 1991; Temperton et al., 2001). The vertical discretization is based on a hybrid sigma-pressure coordinate (Simmons and Burridge, 1981) solved using a finite-element scheme (Untch and Hortal, 2004). Subgrid-scale processes are parameterized in terms of the resolved variables and include
representations of radiative transfer (Morcrette et al., 2008; Iacono et al., 2008; Mlawer et al., 1997), convection (Tiedtke, 1989; Bechtold et al., 2014, 2008), clouds (Tompkins et al., 2007; Tiedtke, 1993; Forbes et al., 2011; Forbes and Tompkins, 2011), turbulent diffusion (Köhler et al., 2011), subgrid-scale orographic drag (Lott and Miller, 1997; Beljaars et al., 2004), and non-orographic gravity wave drag (Orr et al., 2010). Changes to the atmosphere implemented in IFS cycle 43r1 include a number of updates to the cloud and convection parameterizations, some modifications to the coefficients used in land-surface
coupling, and a mass conservation constraint to the Stochastically Perturbed Parameterized Tendencies (SPPT) scheme (Buizza et al., 2017).

All configurations have 91 vertical levels in the atmosphere and use the cubic octahedral reduced Gaussian (Tco) grid such that the shortest wavelengths in spectral space are represented by 4 model grid points. ECMWF-IFS-LR uses the Tco199 grid, which has a grid-point resolution of ∼50 km, and an 1800 s time-step. ECMWF-IFS-LR uses the Tco399 grid, which has a
grid-point resolution of ∼25 km, and an 1200 s time-step. All computations for physical parameterizations are calculated on the atmospheric reduced Gaussian grid. Note that although we use the term 'atmosphere-only' to identify model versions that



are not coupled to a dynamic ocean, such configurations still retain two-way coupling with the land-surface and ocean-wave models.

## 2.2 Land surface model

The land surface component of IFS is the Hydrology Tiled ECMWF Scheme of Surface Exchanges over Land (HTESSEL;
Balsamo et al., 2009), with an updated snow scheme (Dutra et al., 2010) and carbon module (CHTESSEL; Boussetta et al., 2013b). Each model grid box is divided into nine 'tiles': two vegetated fractions (high and low vegetation without snow), one bare soil fraction, three snow/ice fractions (snow on bare ground/low vegetation, high vegetation with snow beneath, and sea-ice, respectively), and three water fractions (interception reservoir, ocean, lakes). Grid box surface fluxes are calculated separately for the different subgrid surface fractions leading to a separate solution of the surface energy balance equation and
skin temperature for each of these tiles.

Vegetation types are derived from the Global Land Cover Characteristics data set (GLCC; Loveland et al., 2000), which provides a 1 km resolution classification of 20 land surface types based on the Biosphere-Atmosphere Transfer Scheme (BATS; Dickinson et al., 1993). Seasonal variation in vegetation is included by prescribing a monthly leaf area index climatology based on MODIS satellite data (Boussetta et al., 2013a). The snow scheme is an energy- and mass-balance model that represents an
additional layer on top of the upper soil layer, with independent prognostic thermal and mass contents (Dutra et al., 2010).

## 2.3 Wave model

The ocean-wave model is an evolved version of WAM (Komen et al., 1994) and solves a wave energy balance equation to determine the evolution of a two-dimensional wave spectrum that provides a statistical description of ocean waves over different frequencies and propagation directions (Janssen, 2004). The ocean-wave model interacts with the atmosphere and oceans
through exchange of the Charnock parameter that determines the sea surface roughness (Janssen, 2004) and by introducing a sea-state depending modification of the turbulent kinetic energy flux used for ocean mixing (Janssen et al., 2013; Breivik et al., 2015). The ocean-wave model is run on an irregular latitude-longitude grid with a resolution of approximately 1 degree in ECMWF-IFS-LR and 0.5 degrees in ECMWF-IFS-HR.

## 2.4 Ocean and sea-ice models

The ocean component of ECMWF-IFS is based on version 3.4 of the Nucleus for European Models of the Ocean (NEMO), which consists of a hydrostatic, finite-difference, primitive equation general circulation model (Madec, 2008) coupled to version 2 of the Louvain-la-Neuve Sea-Ice Model (LIM2; Bouillon et al., 2009; Fichefet and Maqueda, 1997). NEMO v3.4 uses an energy and enstrophy conserving vector-form of momentum advection and has a linear free-surface with semi-implicit time-stepping for the hydrostatic pressure gradient solver. The vertical discretization is a z-coordinate with 75 levels and partial cell
thicknesses at the sea floor. The vertical mixing of tracers and momentum is parameterized using the turbulent kinetic energy (TKE) closure scheme (Gaspar et al., 1990).



ECMWF-IFS-HR uses the ECMWF operational configuration of NEMO, which is based on the 'ORCA025' eddy-permitting tripolar grid with a nominal resolution of ∼0.25 degrees. The NEMO configuration used in ECMWF-IFS-LR is not used operationally at ECMWF and is based on the 'ORCA1' tripolar grid, which has a nominal horizontal resolution of ∼1 degree and meridional refinement to ∼0.3 degrees near the equator. The ECMWF-IFS-LR version of NEMO is configured to be as

close as possible to the ECMWF-IFS-HR version of NEMO with differences limited to resolution-dependent parameterizations (table 1). Of particular note is the Gent and Mcwilliams (1990) parameterization for the effect of mesoscale eddies, which is switched on in ECMWF-IFS-LR configuration but switched off in ECMWF-IFS-HR·

The LIM2 sea-ice model shares a horizontal grid with NEMO and is comprised of a three-layer thermodynamic model for the vertical conduction of heat (Fichefet and Maqueda, 1997) and a two-category (solid ice and leads) representation of

dynamics based on the approach of Hibler (1979). Ice motion is driven by the ocean currents and surface wind stress and we adopt the viscous-plastic rheology used in the operational implementation of LIM2 at ECMWF. For the computation of freshwater exchanges, sea ice is assumed to have a constant reference salinity of 6 PSU.

## 2.5 Coupling

Energy, mass, momentum, and turbulent kinetic energy fluxes are exchanged between sub-models with a coupling frequency of

1 hour in all configurations. Coupling is achieved using the sequential single-executable strategy described by Mogensen et al. (2012). One coupled interaction that is not represented is the link between precipitation over land and runoff into the ocean. To mitigate the impact of this missing process, a climatological estimate of riverine input is applied over coastal grid points and the globally integrated freshwater input to the ocean is set to zero at each time step to prevent unwanted drifts in ocean salinity.

An important difference between the experiments described here and the operational configuration of IFS cycle 43r1 is

the treatment of sea-ice coupling. In operational forecasts, prognostic sea-ice concentrations from LIM2 are coupled with the atmosphere model, but sea-ice skin temperatures are calculated using the IFS land-surface thermodynamics combined with an assumed ice thickness and an observationally-derived sea-ice albedo climatology. Early experiments with a prototype version of ECMWF-IFS suffered from extreme biases in Arctic sea-ice volume. To alleviate, but unfortunately not resolve, these biases, we deviate from the operational configuration and couple skin temperatures from LIM2 and disable the IFS land-

surface thermodynamics over sea-ice. The IFS sea-ice albedo climatology was retained as coupling the prognostic albedo from LIM2 exacerbated existing biases. This result can be explained in part by the absence of melt pond processes in LIM2, which causes prognostic albedo values to be much higher than observed during the summer.

## 2.6 External climate forcings

The external radiative forcings used for transient historical experiments follow CMIP6 recommendations (CMIP6 website,

2017). Greenhouse gas concentrations are specified using the data set described in Meinshausen et al. (2017). Concentrations of $CO_2$, $CH_4$, $N_2O$, and $CFC_{12}$ are explicitly prescribed and the combined effect of other greenhouse gas species is included as an effective concentration of $CFC_{11}$. Time-varying historical ozone concentrations are specified using monthly three-dimensional ozone distributions from the International Global Atmospheric Chemistry/Stratosphere-troposphere Processes And their Role




in Climate Chemistry-Climate Model Initiative ozone database (IGAC/SPARC CCMI ozone data for CMIP6, 2017). Solar forcing is applied as annual means of total solar irradiance derived from the CMIP6 solar forcing described in Matthes et al. (2017).

Tropospheric aerosol forcing is specified using version 2 of the Max-Plank Institute Aerosol Climatology Simple Plume model (MACv2-SP Stevens et al., 2017), which was implemented within the IFS radiation scheme specifically for this configuration. MACv2-SP enables the prescription of the anthropogenic aerosol optical properties and associated Twomey effect and consists of nine spatial plumes related to the major anthropogenic sources. The amplitudes of each plume are scaled through time to provide region-specific variations in tropospheric aerosol through the historical period.

Volcanic forcing is specified as a total stratospheric aerosol optical depth (SAOD) that varies with time and latitude. SAOD is derived from an offline vertical integration of extinction coefficients at 550 nm from a simplified version of the CMIP6 stratospheric aerosol data set v2 (Stratospheric aerosol data for CMIP6, 2017). Extinction coefficients below the observed climatological tropopause (we use the 'dynamic' tropopause of Wilcox et al., 2012) do not contribute to estimates of SAOD.

In atmosphere-only experiments, daily mean SSTs and sea-ice concentrations are specified using a $0.25 \times 0.25$ degree version of the observation-based HadISST2 data set (Rayner et al., 2016; Titchner and Rayner, 2014). The climate forcing from changes in land-use is not prescribed.

## 2.7 Tuning

Approaches to climate model 'tuning' vary depending on the intended scientific purpose of the model and core mission of the centre(s) responsible for model development (Hourdin et al., 2017; Schmidt et al., 2017). The aim of ECMWF-IFS was not to produce the best possible model for climate applications, but to provide a configuration that was traceable to a recent operational forecast model and also sufficiently stable to run multi-decadal experiments in coupled mode. Under this constraint, tuning was limited to several cases where leaving the model unchanged would have introduced intolerable drifts in the climate system.

The principal tuning target was global surface energy balance in atmosphere-only simulations over the period 2005-2013. Simulations were considered adequate if the global average heat flux into the ocean (relative to Earth's surface area) was within the observed range of 0.2-1.0 W/m$^2$ (Wild et al., 2013). It was found that net surface energy balance was strongly sensitive to atmospheric resolution such that the net surface heating in ECMWF-IFS-HRA was about 1 W/m$^2$ less than in ECMWF-IFS-LRA. To account for this difference, it was necessary to tune the cloud properties in ECMWF-IFS-HR by reducing the autoconversion threshold for liquid precipitation over the ocean. The only other adjustment to the atmospheric model relative to the operational implementation of cycle 43r1 was to modify the non-orographic gravity wave drag in all configurations to improve the representation of the quasi-biennial oscillation. This change was made following recommendations from the ECMWF seasonal forecast team and will be implemented operationally in future IFS cycles.

Only one element of the system was tuned in coupled mode. Early prototype experiments suffered from a severe and monotonic trend in Arctic sea-ice volume that was not addressed by modifications to the air-ice-sea coupling. To stabilize this bias it was necessary to scale the climatological sea-ice albedos used in calculations of sea-ice skin temperature by a constant factor





of 0.95 across all spectral bands. This scaling was a pragmatic approach to partially mitigate the underlying errors in the downward radiation over the Arctic and the response of LIM2 sea ice model. The correction is only applied to coupled integrations and atmosphere-only integrations use the original IFS albedo climatology.

## 2.8 HighResMIP experiments

This section describes atmosphere-only and coupled experiments following the CMIP6 HighResMIP protocol (Haarsma et al., 2016) that were performed with ECMWF-IFS under the auspices of the PRIMAVERA project. A full list of experiments and their associated ECMWF run identifiers are provided as an appendix.

### 2.8.1 highresSST-present

The *highresSST-present* ensemble covers the period 1950-2014 and consists of 6 members from ECMWF-IFS-LRA and 4
members from ECMWF-IFS-HRA. These atmosphere-only integrations are forced with observed SSTs, observed sea-ice concentrations, and external radiative forcings as described in section 2.6. Atmosphere and land-surface models are initialized with conditions representative of January 1 1950 using data from the ERA-20C reanalysis (Poli et al., 2016). Ensemble members are distinguished by different seeds to the SPPT scheme.

### 2.8.2 spinup-1950

To provide initial conditions for the coupled experiments described below, 50 year spin-up integrations are performed with ECMWF-IFS-LR, ECMWF-IFS-MR, and ECMWF-IFS-HR using external forcings fixed at values from the year 1950. The ocean is initialized from rest using a temperature and salinity climatology representative of 1950s derived from the EN4 objective analysis (Good et al., 2013). In regions with initial sea-surface temperatures below freezing, sea-ice is initialized from rest with a uniform thickness of 3 m in the Arctic and 1 m in the Antarctic. The atmosphere and land-surface models are
initialized in the same way as *highresSST-present*.

### 2.8.3 hist-1950

The coupled *hist-1950* ensemble covers the period 1950-2014 and consists of 6 members from ECMWF-IFS-LR and 4 members from ECMWF-IFS-HR, and a single member from ECMWF-IFS-MR. Experiments are initialized from the end of *spinup-1950* and time-varying external forcings are specified using the data sets described in section 2.6. As in *highresSST-present*,
ensemble members are distinguished by different seeds to the atmospheric stochastic physics. Since some ocean properties have significant persistence on multi-annual time-scales, members of *hist-1950* are not completely independent for the first few years of the experiment. However, preliminary analysis of North Atlantic ocean heat content indicates that the ensemble diverges rapidly within the first few years with ensemble spread saturated after about 10-15 years.



### 2.8.4 control-1950

The *control-1950* experiments consist of single-member 100 year integrations initialized from the end of the associated *spinup-1950* experiments with forcings fixed at 1950 values. These experiments run in parallel to *hist-1950* and are useful to identify long-term trends that are unrelated to changes in radiative forcings.

## 3   Model evaluation

### 3.1   Global temperature trends and the planetary energy budget

#### 3.1.1   Surface temperature

Observed long-term trends in global 2m temperature over land ($T_{2m}$) and global sea surface temperature (SST) are well-reproduced in ECMWF-IFS (figure 1a-b) with only minor differences between low- and high-resolution configurations. The global surface temperature response in ECMWF-IFS-LRA and ECMWF-IFS-HRA is tightly constrained by the prescription of observed SSTs. In contrast, ECMWF-IFS-LR and ECMWF-IFS-HR have no such constraint and no attempt was made to tune the model to reproduce historical changes in global surface temperature. The long term trends in global surface temperature are thus an emergent property of the imposed radiative forcings and coupled model, and the agreement with observations gives us confidence in the utility of ECMWF-IFS for multi-decadal climate applications.

On shorter time-scales, both coupled and atmosphere-only configurations accurately simulate the transient cooling over land associated with large volcanic eruptions, although the SST response to volcanic eruptions in ECMWF-IFS-LR and ECMWF-IFS-HR is generally too large compared to observations. Over the so-called 'hiatus' period of the early 21st century, global SST anomalies in ECMWF-IFS-LR and ECMWF-IFS-HR increase faster than observed, particularly when compared to an earlier version of the HadISST data set (Rayner et al., 2003) rather than the HadISST2 forcing data set recommended by HighResMIP. The discrepancy in the rate of warming between models and observations during this period is also common in CMIP5 models and has been attributed to differences in the phase of observed and simulated modes of multi-decadal variability (e.g. Meehl et al., 2013; Roberts et al., 2015).

#### 3.1.2   Planetary energy budget

Table 2 shows radiative and turbulent components of the global mean net surface heat flux ($F_{sfc}$) for the period 2000-2014 in coupled and atmosphere-only configurations. The individual components are well within observational uncertainty estimates (Wild et al., 2013) in all configurations. $F_{sfc}$ is in good agreement with observations in atmosphere-only simulations and ECMWF-IFS-HR, and slightly too high in ECMWF-IFS-LR. This offset in $F_{sfc}$ between experiments is a result of the time-evolving SST biases that develop in coupled integrations (see section 3.3) combined with the short duration of the spinup experiment (50 years) relative to the equilibration time of the ocean interior (hundreds of years).



Changes in global mean upper 700 m ocean temperature ($T_{0-700m}$) from coupled experiments are shown in figure 1c against the range of estimates from the observation-based ORAS4 ensemble (Balmaseda et al., 2013). Consistent with the values of $F_{sfc}$ in table 2, changes in $T_{0-700m}$ are generally too large compared to observations, particularly in ECMWF-IFS-LR. The anomalously high rate of ocean heat uptake in the coupled experiments associated with temperature 'drift' in the ocean interior

that occurs in response to the development of surface ocean biases. It is not practical to run high-resolution coupled experiments for the many hundreds of years required for the ocean interior to reach equilibrium with the imposed forcings. Instead, the impact of the time-evolving forcing on $T_{0-700m}$ can be evaluated by expressing changes in *hist-1950* with respect to changes in *control-1950*. Following this adjustment (figure 1c), estimates of $T_{0-700m}$ in ECMWF-IFS-LR and ECMWF-IFS-HR are in good agreement with one another, despite the differences in long-term drift, and very close to the range of estimates from

ORAS4. In addition, both adjusted and unadjusted estimates of $T_{0-700m}$ contain realistic cooling signals associated with major volcanic eruptions that are also present in ORAS4.

### 3.1.3 Conservation in the atmosphere

Due to the limited heat capacity of the atmosphere, the net radiation flux at the top of the atmosphere ($F_{toa}$) should almost exactly balance $F_{sfc}$ on time-scales of a year or more (Palmer and McNeall, 2014). Despite the favourable comparisons with

various aspects of the global energy budget (figures 1a-c, table 2), annual mean values of $F_{sfc}$ and $F_{toa}$ do not exactly balance one another in ECMWF-IFS, indicating a spurious source of energy in the atmosphere of approximately 1 W/m$^2$. The absence of closure for atmospheric energy budgets has been described previously for CMIP5 models (Hobbs et al., 2016) and an earlier version of the IFS (Hersbach et al., 2015). Providing such 'energy leaks' are time-constant and independent of forcings, results can be interpreted in an anomaly framework without biasing results (Hobbs et al., 2016).

The magnitude of non-conservation in ECMWF-IFS is resolution dependent with values of about 1.2 W/m$^2$ and 1.0 W/m$^2$ at Tco199 and Tco399, respectively. Further investigation revealed that the leading contribution to non-conservation was the non-conservation of moisture during the semi-Lagrangian advection. In additional test experiments, application of a mass-fixer algorithm that preserves the global integrated moisture mass before and after calls to the advection (Diamantakis and Flemming, 2014; Bermejo and Conde, 2002) significantly improved conservation of moisture and heat in the atmosphere and

should be used in future climate experiments with ECMWF-IFS. Despite these issues, anomalies in $F_{toa}$ are sufficiently well correlated with anomalies in $F_{toa}$ (correlations are 0.98 and 0.97 in ECMWF-IFS-LR and ECMWF-IFS-HR, respectively) that it is possible to interpret the resulting climate in a physically meaningful way. However, one important caveat is that it is necessary to consider $F_{sfc}$ rather than $F_{toa}$ when evaluating planetary energy balance. This is particularly important if tuning the energy balance in an atmosphere-only configuration for use in the coupled system as it is mostly through adjustments in SST,

and hence changes in $F_{sfc}$, that the climate system responds to an imposed forcing.





### 3.2 Biases in the atmosphere

#### 3.2.1 Near-surface temperature biases

The near-surface air temperature over land ($T_{2m}$) in ECMWF-IFS is generally cooler than observed (figure 2), with all configurations exhibiting particularly severe cold biases in regions of high elevation and a localized warm bias in eastern Siberia. Increasing atmospheric resolution from 50 km to 25 km in atmosphere-only experiments (figures 2a-b) has very little impact on $T_{2m}$ biases. However, coupling to the NEMO ocean model increases the magnitude of the northern hemisphere cold bias (figures 2c-d) because of the introduction of a cold bias in North Atlantic SSTs that is particularly severe in ECMWF-IFS-LR (figures 3a,c). Despite performing better than ECMWF-IFS-LR in the northern hemisphere, ECMWF-IFS-HR suffers from an increased warm bias over the Australian continent due to a warm bias in the Southern Ocean that is intensified with the higher resolution ocean (figures 3a,c). The origins of SST biases in coupled configurations are discussed in more detail in section 3.3.

#### 3.2.2 Precipitation biases

Atmosphere-only configurations of ECMWF-IFS suffer from a characteristic pattern of excessive precipitation over the tropical oceans that is insensitive to changes in atmospheric resolution (figures 4a-b). Although the zonal mean structure of tropical precipitation biases is similar in coupled and atmosphere-only configurations (figure 4e), the presence of spatially varying SST biases in the coupled system (figures 3a,c) results in several basin-specific differences in tropical precipitation relative to the atmosphere-only configurations. In particular, both coupled configurations suffer from an intensification of precipitation biases in the off-equatorial tropical Pacific and western tropical Indian Oceans and a reduction of precipitation over the equatorial Pacific. In the case of ECMWF-IFS-LR, an equatorial Pacific cold SST bias (figures 3a,c) results in a precipitation deficit . These differences are indicative of the impact of coupling on the large scale circulation of the tropical atmosphere. In both ECMWF-IFS-LR and ECMWF-IFS-HR, the cold SST bias in the North Atlantic (figures 3a,c) drives a southward shift of the Atlantic Intertropical Convergence zone (ITCZ) and reduced precipitation over the southern edge of west Africa. This bias is larger in ECMWF-IFS-LR reflecting the larger bias in North Atlantic SSTs.

The off-equatorial Pacific rain bands are a typical feature of coupled climate models and well-known as the 'double-ITCZ problem' (e.g. Li and Xie, 2014; Adam et al., 2017). The double-ITCZ bias in ECMWF-IFS-LR and ECMWF-IFS-HR is largely confined to the west Pacific and is generally an improvement on the previous generation of CMIP models where the southern rain band is much stronger (see figure 1a of Adam et al., 2016). Following Adam et al. (2016), we calculate the tropical precipitation asymmetry index $A_p$ defined as

$$A_p = \frac{\bar{P}_{0-20^\circ N} - \bar{P}_{20^\circ S-0}}{\bar{P}_{20^\circ S-20^\circ N}} \tag{1}$$

where over-bars denote the zonal average over the Pacific sector and subscripts indicate the area-weighted meridional average between the specified latitudes. The observation-based value of $A_p$ is 0.19 and can be compared to values of 0.11 in ECMWF-





IFS-LR and 0.15 in ECMWF-IFS-HR, the latter of which is better than 26 of the 28 CMIP models compared in Adam et al. (2017).

### 3.2.3 Cloud radiative forcing biases

To further understand the biases in $T_{2m}$, we consider biases in the surface radiation budget that result from errors in cloud
radiative forcing (CRF; figures 5-7). CRF is given by the following equation

$$CRF = F - F_{clearsky} \tag{2}$$

where F is a downward radiative flux at the surface and 'clear sky' indicates the radiative fluxes that would be received in the absence of clouds. Spatial variations in the sign of the total CRF bias largely reflect errors in short-wave CRF (figure 6) that are partially compensated by opposing errors in long-wave CRF (figure 7). Total CRF biases over the ocean are generally negative,
except in the Southern Ocean and regions of upwelling. CRF biases over land have more small scale structure due to variations in topography and land-surface characteristics. The large-scale spatial structure of CRF biases are set by the atmospheric model and are relatively insensitive to a change in atmospheric resolution from 50 km to 25 km. CRF biases are modified by ocean coupling in regions where cloud properties are sensitive to SST biases, either due to changes in the large-scale atmospheric circulation or because of cloud-SST-radiation feedbacks. For example, ocean coupling modifies the position and magnitude
of the short-wave CRF bias in the equatorial oceans surrounding the maritime continent and increases the magnitude of the negative short-wave (and positive short-wave) CRF biases in the sub-tropical oceans.

Over land, there is no simple causal relationship between $T_{2m}$ biases and total CRF biases (figure 5). For example, all model configurations exhibit a strong negative $T_{2m}$ bias over the Tibetan Plateau that occurs despite positive biases in SW CRF and total CRF (figure 6). To better understand such discrepancies, we also consider the impact of biases in surface albedo on the
net short-wave radiation at the surface (figures 8 and figure 9). Over the oceans, biases in surface albedo are a consequence of biases in sea-ice cover and/or its assumed albedo. For example, the loss of Antarctic sea ice in coupled configurations results in a strong reduction of surface albedo in the Southern Ocean (figures 8c-d) and an excess of absorbed short-wave radiation in this region (figures 9c-d). Over land, biases in surface albedo reflect biases in snow cover and/or inaccurate specification of the land-surface characteristics. In some regions over land (e.g. northern Africa, coastal Greenland, northern Siberia, and
the Tibetan Plateau), the cold biases in $T_{2m}$ can be explained by positive biases in surface albedo that result in insufficient net surface short-wave radiation. In other locations, such as the mountain chains of South America, near-surface cold biases are associated with positive biases in net surface short-wave radiation suggesting a dynamic origin for temperature errors in these locations.

### 3.2.4 Upper atmosphere biases

In ECMWF-IFS-LRA and ECMWF-IFS-HRA, lower troposphere temperature biases (figures 10) are relatively small. In ECMWF-IFS-LR and ECMWF-IFS-HR, lower troposphere temperature biases, and associated changes to the westerly winds, reflect SST biases in the North Atlantic and the Southern Ocean (figures 3a,c). All configurations suffer from a prominent




cold bias in the lower stratosphere that is increased in magnitude at higher horizontal resolution. This bias is not yet fully understood, but is thought to be a consequence of spurious mixing across the tropopause associated with small scale variability that is intensified at horizontal higher resolutions with the cubic octahedral grid and improved by increased vertical resolution (personal communication, Tim Stockdale). In the upper stratosphere, all configurations exhibit a warm bias that is maximal in

the mid-latitudes of the winter hemisphere (figure 10) and relatively insensitive to changes in atmospheric resolution.

    Zonal mean westerly wind biases are dominated by errors in the position and/or intensity of jets in the upper troposphere and stratosphere (figures 11) and reflect the meridional structure of temperature biases (figures 10). The largest bias in westerly winds is common to all configurations and is located in the tropical stratosphere between 5 and 50 hPa. This bias has been identified in previous versions of the IFS and is known to be sensitive to the details of vertical diffusion in the atmosphere

(Hersbach et al., 2015). In accordance with the thermal-wind relationship, the vertical shear and absolute magnitude of this bias is slightly higher in ECMWF-IFS-HR and ECMWF-IFS-HRA because of the larger magnitude temperature bias in the lower stratosphere. In ECMWF-IFS-LR and ECMWF-IFS-HR, SST biases drive a northward shift of surface winds in the region 60°S-30°S and an intensification of the southern hemisphere polar stratospheric vortex. This effect is most prominent in ECMWF-IFS-HR due to the larger magnitude Southern Ocean warm bias.

**3.3   Biases in the ocean and cryosphere**

This section presents an overview of the climatological biases in the ocean and cryosphere in coupled configurations of ECMWF-IFS, including selected results from the 'mixed resolution' configuration ECMWF-IFS-MR, which combines the atmosphere from ECMWF-IFS-LR with the high resolution ocean from ECMWF-IFS-HR. A number of issues are common to all three coupled configurations, including negative SST biases in the North Atlantic, positive SST biases in the Southern

Ocean, excessive Arctic sea-ice, positive SSS biases in the Arctic, negative SSS biases in the tropical South Pacific associated with the double ITCZ, insufficient mixing in the Southern Ocean, and inadequate heat loss over the Gulf Stream extension (figure 3). It is also clear from figure 3 that differences in the mean state of the coupled system are dominated by the change in ocean resolution. Indeed, the ocean biases in ECMWF-IFS-MR and ECMWF-IFS-HR are nearly identical despite the doubling of atmospheric resolution from ∼50 km to ∼25 km (figure 3).

With an obvious exception of the Southern Ocean, many biases show a clear improvement with increased horizontal resolution in the ocean. For example, the equatorial Pacific and North Atlantic are two key locations where the time-varying 'eddy' heat transports are better resolved in ECMWF-IFS-MR/-HR (figures 12a-b) leading to changes in the ocean heat budget and and improved SSTs (figure 3a-c). Conversely, some ocean biases are common to all three coupled configurations suggesting either a structural error that is present in both high and low resolution versions of NEMO or common errors in the forcing from

the atmospheric model. For example, the mixed layer bias in the Southern Ocean is common to all three models despite rather different biases in SST and wind stress (not shown). This suggests a deficiency in the TKE scheme for vertical mixing that is common to all three coupled models.

    Given that the ECMWF coupled model is used principally for operational applications with lead-times much shorter than one year, it is also interesting to consider the time-scale-dependence of ocean biases. In short (< 1 year) integrations, coupled





model biases reflect the influence of 'fast' processes combined with impact of the ocean initialization strategy. On decadal and longer time-scales, coupled model biases include the impacts of 'slower' coupled processes and are more representative of the asymptotic behaviour of the model. Some elements of the coupled system respond relatively quickly such that biases evident in the first year (figure 13) are remarkably similar to biases calculated using 50 years of data (figure 3). For instance, biases

in Southern Ocean mixed layer depth show very little change after the first year of integration highlighting the dominant role of 'fast' processes in determining these changes. In contrast, and despite the fundamental differences in the representation of the underlying ocean dynamics, it is more difficult to discriminate between North Atlantic SST and surface heat flux biases in high and low resolution configurations using only a single year of data (figure 13). These comparisons emphasize that multi-decadal climate experiments can provide information for the coupled model development process that is complementary to that

available from short integrations and (re)forecast data sets.

### 3.3.1 North Atlantic

The negative North Atlantic SST bias is present in all coupled configurations, but is particularly severe in ECMWF-IFS-LR. This bias is a consequence of inadequate northward ocean heat transport (figure 12a) and a sluggish Atlantic meridional over-turning circulation (AMOC; figure 12b). Atlantic ocean heat transports are higher in ECMWF-IFS-MR and ECMWF-IFS-HR,

but still lower than estimates derived from hydrographic sections (Ganachaud and Wunsch, 2003). In contrast, the northward heat transports in the Indo-Pacific are similar in all coupled configurations and in reasonable agreement with observational estimates (figure 12b).

Comparison of simulated AMOC stream function profiles with observations from the RAPID-MOCHA array at 26 °N (McCarthy et al., 2015) demonstrates that all configurations underestimate the maximum strength of the AMOC and the depth

of the southward return flow, although ECMWF-IFS-MR/-HR performs substantially better than ECMWF-IFS-LR (figure 12c). Previous work has demonstrated that the agreement between observed and simulated stream functions at 26 °N can be improved by accounting for a sensitivity to the choice of reference level used in the geostrophic approximation employed in the RAPID-MOCHA observations (Roberts et al., 2013). However, even when this sensitivity is taken into account, there is a clear bias towards shallower levels in the simulated overturning cells (figure 12c).

Figure 12d shows variations in the strength of the AMOC at 26 °N against variations in overturning and horizontal gyre contributions to ocean heat transport (see Johns et al., 2011, for details of this decomposition). From this plot it is clear that most of the differences in heat transport between coupled models and observations can be explained by differences in the strength of the overturning circulation. In addition, the model heat transports for the same AMOC strength are offset relative to the observations by about ∼0.2 PW. This is due to inaccuracies in the distribution of volume transports in temperature classes

due to biases in the depth-structure of the overturning and the near-surface temperature biases. ECMWF-IFS-LR also exhibits a spurious heat transport compensation such that increased heat transport by the overturning circulation is offset by a reduced heat transport by the gyre. This relationship is not evident in the RAPID-MOCHA observations or ECMWF-IFS-MR/-HR and is likely related to the absence of the Florida Straits, and its associated depth-confined flow, in the ORCA1 NEMO model.





### 3.3.2 Southern Ocean

The positive SST bias in the Southern Ocean (figure 3a-c) is most evident in ECMWF-IFS-MR/-HR and is associated with reduced Antarctic sea-ice cover (figure 14), a shallow bias in the depth of the ocean mixed layer (figure 3g-i), and increased sea surface height (SSH) around Antarctica (figure 3m-o). In ECMWF-IFS-MR/-HR, these biases in the surface ocean are

associated with a weak Antarctic Circumpolar Current (ACC). For example, mean volume transports through the Drake Passage are $\sim$60 Sverdrups (1 Sv = $10^6$ m$^3$/s) in ECMWF-IFS-MR/-HR, compared to $\sim$115 Sv in ECMWF-IFS-LR and $\sim$130 Sv from observations (Cunningham et al., 2003; Whitworth III, 1983).

    One of the drivers of the Southern Ocean SST bias is a 10-20 W/m$^2$ bias in short-wave cloud radiative forcing that is insensitive to resolution and present in both coupled and atmosphere-only configurations (figure 6). This bias in downward

short-wave radiation at the surface is reinforced by a coupled sea-ice albedo feedback that is particularly strong in ECMWF-IFS-MR and ECMWF-IFS-HR (figures 8 and 9). This result is consistent with previous studies that have documented the tendency for atmospheric models to underestimate the albedo of clouds over the Southern Ocean (Bodas-Salcedo et al., 2012, 2014) and the subsequent impact on Southern Ocean SSTs when coupling to the NEMO ocean model (e.g. Williams et al., 2015).

The different responses in the Southern Ocean in ECMWF-IFS-LR and ECMWF-IFS-MR/-HR are likely a consequence of the different representations of the ocean mesoscale. Specifically, the balance between wind-driven and baroclinic-instability-driven circulations is parameterized in ECMWF-IFS-LR but not in ECMWF-IFS-MR/-HR. Although the ORCA025 version of the NEMO ocean model used in ECMWF-IFS-MR/-HR is considered 'eddy-permitting', it is far from fully resolving the energetic mesoscale eddy field that extends along the path of the ACC (Marshall and Speer, 2012). Recent work by Hewitt

et al. (2016) found that comparable Southern Ocean biases in the HadGEM3 climate model were reduced when increasing the horizontal resolution of NEMO from 1/4° to 1/12°.

### 3.3.3 Sea ice biases

Simulated and observation-based climatologies of sea-ice area and volume are shown in figure 14a-d. In the northern hemisphere, ECMWF-IFS-MR/-HR effectively captures the seasonal cycle of sea-ice area, though the extent is too high during the

September minimum. In contrast, ECMWF-IFS-LR suffers from an excess of northern hemisphere sea-ice in all seasons. During the maximum extent, the bias in ECMWF-IFS-LR extends over the Labrador sea and parts of the western North Atlantic sub-polar gyre leading to suppressed ocean convection and a positive feedback on the AMOC and North Atlantic SST biases. The increased transport of freshwater by sea ice also reduces the salinity of the North Atlantic sub-polar gyre, which further contributes ro the reduced AMOC in ECMWF-IFS-LR.

Northern hemisphere sea-ice is too thick in all three coupled configurations. The resulting biases in ice volume are most severe in ECMWF-IFS-LR (figure 14b) due to the combined effect of biases in extent and thickness. In part, these biases can be explained by excessive ice growth in response to negative biases in both long-wave and short-wave CRF over the Arctic (figures 6 and 7).



In the southern hemisphere, ECMWF-IFS-MR/-HR has too little sea-ice during the September maximum and all three coupled configurations have too little sea-ice during the February minimum. These deficiencies in southern hemisphere sea-ice extent are a consequence of the previously discussed biases in downward short-wave radiation and the resulting warm bias in Southern Ocean SSTs that is particularly extreme in ECMWF-IFS-MR/-HR.

## 4   ENSO variability and teleconnections

The El Niño-Southern Oscillation (ENSO) is the leading mode of tropical ocean-atmosphere variability (McPhaden et al., 2006). Tropical SST anomalies associated with ENSO can also influence the extra-tropical atmosphere through their impact on tropical convection and patterns of upper-atmosphere convergence/divergence that act as a source of Rossby waves, which subsequently propagate and dissipate at higher latitudes (Hoskins and Karoly, 1981; Trenberth et al., 1998; Simmons et al., 1983; Held et al., 1989). Accordingly, models such as the IFS that are used to make seasonal forecasts need to accurately represent the ocean-atmosphere interactions that drive ENSO variability and the teleconnections that communicate its influence around the globe (Latif et al., 1998). The following section provides an assessment of ENSO variability in ECMWF-IFS and includes discussion of the important ocean-atmosphere feedbacks and global teleconnection patterns.

### 4.1   Variability of ENSO indices

Figure 15 shows times series of Niño3.4 SST anomalies from HadISST2 compared with four members of hist-1950 from each of ECMWF-IFS-LR and ECMWF-IFS-HR. El Niño and La Niño events are highlighted in red and blue, respectively, along with the total number of events in the period 1950-2014. When all ensemble members and experiments are considered together, the variance of detrended monthly Niño3.4 SST values is higher in ECMWF-IFS-HR ($0.84$ K$^2$) than in ECMWF-IFS-LR ($0.68$ K$^2$) and in better agreement with observed values ($0.78$ K$^2$).

Previous generations of coupled climate models have struggled to simulate the asymmetry in ENSO variability that is characterized by shorter more intense El Niño events and longer less intense La Niña events (Zhang and Sun, 2014). This asymmetry is particularly evident in the eastern Pacific where it manifests as a positive skewness in the Niño1+2 SST index (Burgers and Stephenson, 1999). ECMWF-IFS-HR and ECMWF-IFS-LR both correctly simulate the sign of ENSO asymmetry in the Niño1+2 SST index (figure 16a). The magnitude of the asymmetry is generally better in ECMWF-IFS-HR (skewness=0.53) than in ECMWF-IFS-LR (skewness=0.31) compared to observations (skewness=1.46), though both models miss the most extreme positive SST events in the Niño1+2 region (figure 16a)

To further characterize the periodicity and asymmetry of ENSO variability, we calculate the frequency of warm and cool events as a function of duration (figures 16b-c). Consistent with our conclusions from figure 16a, the distributions of warm and cool event durations are more symmetric in ECMWF-IFS-LR and ECMWF-IFS-HR than observed. For warm events, there is a bias towards too many short ($< 6$ months) events and not enough events lasting 18-24 months. For cool events, both coupled models underestimate the number of events lasting 36-42 months. Given these biases, we might expect that seasonal forecasts



performed with coupled versions of the IFS to have a tendency to underestimate the persistence of long-lasting warm and cool events in the tropical Pacific.

## 4.2 ENSO ocean-atmosphere feedbacks

For models to accurately simulate the observed asymmetry of ENSO variability, they must adequately capture the non-linear

aspects of ENSO dynamics and thermodynamics. However, previous work has demonstrated that many coupled climate models struggle to simulate non-linearities in SST-radiation feedbacks as diagnosed by the relationship between the Niño3.4 SST index and top-of-atmosphere radiation fluxes over the eastern equatorial Pacific (Bellenger et al., 2014; Mayer et al., 2016). These non-linearities are evident from observations during warm events as compensating changes in outgoing long-wave radiation (OLR) and absorbed solar radiation (ASR) (figures 17a-b) associated with the eastward extension of positive SST anomalies

and the onset of deep convection in the eastern Pacific. Previous studies have identified a wide range of model behaviours in this region, which can be related to the differences in the mean climate of the eastern equatorial Pacific and the associated SST and cloud biases (Bellenger et al., 2014; Mayer et al., 2016).

Atmosphere-only configurations of ECMWF-IFS (figures 17c-f) capture this non-linear behaviour, but the compensation between OLR and ASR components is only partial and the negative OLR response to warm events is larger than observed by

satellite radiation data (black crosses in figure 17a) and more similar to the OLR response in the ERA-interim reanalysis (grey crosses in figure 17a). The degree of non-linearity shows no sensitivity to the change in atmospheric horizontal resolution from 50 km to 25 km. These differences indicate a deficiency in the atmospheric model response to observed SST variability, which is likely related to the radiative impact of convective clouds in the eastern equatorial Pacific.

In the coupled configurations (figures 17g-j), ECMWF-IFS-LR shows limited evidence for non-linear SST-radiation inter-

actions, whereas the OLR and ASR responses in ECMWF-IFS-HR are more similar to those in the atmosphere-only configurations. Following the arguments of Bellenger et al. (2014) and Mayer et al. (2016), we conclude that the improved ASR and OLR response in ECMWF-IFS-HR is a consequence of the increased ocean horizontal resolution and improvements in the mean state of the equatorial Pacific (see section 3.3). The better representation of non-linear ocean-atmosphere feedbacks in this region may also partly explain the improved asymmetry of ENSO variability in ECMWF-IFS-HR compared to ECMWF-

IFS-LR.

## 4.3 ENSO teleconnections

The local and remote SST response to variations in northern winter (DJF) SST anomalies in the Niño4 region are shown in figure 18. The DJF period is chosen because tropical forcing of the extra-tropics is generally strongest in the northern winter (Trenberth et al., 1998). Both ECMWF-IFS-LR and ECMWF-IFS-HR capture signals in the central and eastern tropical Pacific

and in the Indian Oceans. However, neither model is able to capture the negative correlations north and south of the tropical Pacific that are present in the observations. In ECMWF-IFS-LR, positive correlations extend too far west into the west Pacific warm pool and there is a spuriously strong correlation with SSTs in the North Atlantic sub-polar gyre, which is likely a consequence of the strong cold bias and excessive wintertime sea-ice in this region in ECMWF-IFS-LR.





Figure 19 shows covariance maps for global rainfall anomalies associated with DJF rainfall over the the Niño4 region for both coupled and atmosphere-only experiments. The atmosphere-only configurations successfully capture the pattern of rainfall anomalies over the tropical Pacific, tropical Atlantic, maritime continent, and equatorial South America. However, neither atmosphere-only configuration is able to capture the magnitude of the precipitation responses over the United States,

sub-tropical North Atlantic, and southern half of South America. In addition, both configurations overestimate the magnitude of precipitation anomalies over the Indian Ocean suggesting a common deficiency in the response of the Walker circulation to diabatic forcing in the Niño4 region. Precipitation responses in the coupled configurations behave similarly to the atmosphere-only counterparts. The exception is the clear degradation in the equatorial Pacific in ECMWF-IFS-LR that is related to the previously described issues with the double ITCZ and equatorial cold tongue bias.

To examine the northern hemisphere wintertime response to diabatic forcing in the tropical Pacific, we calculate covariances of DJF geopotential height anomaly at 500 hPa (Z500) with the normalized rainfall anomaly in the Niño4 region (figure 20). In the ERA-interim reanalysis, increased precipitation in the central equatorial Pacific is generally associated with positive Z500 anomalies over Greenland and Canada, and negative Z500 anomalies over Siberia, the north east Pacific, and the north Atlantic. This pattern projects onto the negative phase of the North Atlantic Oscillation (NAO) and the positive phase of the

Pacific/North American Pattern (PNA), though these modes explain only part of the ENSO response (Straus and Shukla, 2002).

The different configurations of ECMWF-IFS show some similarities in the extra-tropical response to Niño4 precipitation variability, including successful simulation of the PNA-component of the response and failure to capture the Siberian low. In contrast, there are clear differences between configurations in the North Atlantic/European sector. In this region, ECMWF-IFS-HR and ECMWF-IFS-HRA more accurately simulate the structure of the Z500 response to Niño4 precipitation, although the

magnitude of the response is still too weak. Only in the coupled version of ECMWF-IFS-HR do Z500 anomalies project onto the negative phase of the NAO. Taken at face value, these results suggest that increased atmospheric resolution and coupling with a dynamic ocean may improve the response of extra-tropical circulation to tropical forcing in the North Atlantic sector. However, recent work has emphasized the difficulty in evaluating ENSO teleconnection patterns due to the limited sample size in both observational and model data sets (Deser et al., 2017). The atmospheric variability over the North Atlantic on sub-

seasonal to decadal scales is also affected by teleconnections originating from the tropical Indian Ocean (Cassou, 2008; Molteni et al., 2015), and a preliminary assessment of these teleconnections in ECMWF-IFS-LR and ECMWF-IFS-HR suggests a less optimistic picture. A more robust attribution analysis is beyond the scope of this paper, and will require careful consideration of both observational uncertainties and the relationship between signals originated in different parts of the tropics.

## 5   Discussion and conclusions

This paper has presented coupled and atmosphere-only climate model configurations of the European Centre for Medium Range Weather Forecasts Integrated Forecast System (ECMWF-IFS) for different combinations of ocean and atmosphere resolutions. In order to provide configurations that are traceable to a recent operational version of the ECMWF weather forecast model (cycle 43R1), model 'tuning' was limited to several instances where leaving the model unchanged would have intro-



duced intolerable drifts in the coupled climate system. Two atmosphere-only configurations (ECMWF-IFS-LRA, ECMWF-IFS-HRA) and three coupled configurations (ECMWF-IFS-LR, ECMWF-IFS-MR, ECMWF-IFS-HR) were used to perform multi-decadal ensemble experiments following the protocols of HighResMIP/CMIP6. The sensitivity to ocean coupling and the impacts of increased atmosphere and ocean resolution are summarized below.

## 5.1 Impacts of coupling

Coupling to the ocean allows the simulation of coupled ocean-atmosphere phenomena, such as ENSO, at the expense of biases in SST and sea-ice distribution, which are the interface for interactions between the ocean and atmosphere. The biases introduced by coupling ECMWF-IFS to the NEMO ocean model are resolution dependent (see section 3.3), but share some general features including cooling of the northern hemisphere, warming of the Southern Ocean (figure 3), and development of a 'double-ITCZ' bias in tropical precipitation (figure 4). In the atmosphere, the impacts of ocean coupling are generally constrained to the troposphere, with stratospheric biases nearly identical in coupled and atmosphere-only integrations (figures 10 and 11). There is tentative evidence that ocean coupling improves teleconnections between tropical Pacific rainfall and geopotential height anomalies in the North Atlantic, but the significance of this change has not yet been assessed (figure 20).

## 5.2 Impact of atmospheric resolution

The climatological surface biases in ECMWF-IFS are relatively insensitive to an increase in atmospheric resolution from ∼50 km to ∼25 km (figures 2-11). The most obvious impact of the increase in atmospheric resolution is an increase in the magnitude of a prominent cold bias in the lower stratosphere and an associated degradation of the equatorial jet (figures 10 and 11). Other processes that are affected by the change in atmospheric resolution include the conservation characteristics of the semi-Lagrangian advection scheme and the net planetary energy balance. Teleconnections between tropical Pacific rainfall and geopotential height anomalies in the North Atlantic are also improved with increased atmospheric resolution, but the significance of this change has not yet been assessed (figure 20). A more thorough assessment of the impact of atmospheric resolution on variability and extremes, which we expect to be more sensitive than mean biases, will be undertaken as part of the PRIMAVERA project.

## 5.3 Impact of ocean resolution

The asymptotic surface biases in coupled configurations of ECMWF-IFS are dominated by the choice of ocean model resolution (figure 3). However, the timescale required to detect the impact of ocean resolution depends on the process and region being considered (figure 13). The positive impacts of an increase in ocean resolution from ∼100 km to ∼25 km are particularly evident in the North Atlantic and Arctic, and include an improved Atlantic meridional overturning circulation, increased meridional ocean heat transports, and more realistic sea-ice cover. In addition, increasing ocean resolution improves the mean state of the equatorial Pacific, leading to improvements to the magnitude and asymmetry of ENSO variability and better representation of non-linear SST-radiation feedbacks. The negative effects of increased ocean resolution include an amplification



of a Southern Ocean warm bias, weakening of the Antarctic circumpolar current, and the dramatic decline of Antarctic sea ice. These effects are likely a consequence of the 'eddy permitting' rather than 'eddy resolving' nature of the ORCA025 ocean configuration in the Southern Ocean and the disabling of the Gent and Mcwilliams (1990) parameterization.

To give some context to the biases in ECMWF-IFS-LR, ECMWF-IFS-MR, and ECMWF-IFS-HR, we compare with the
range of zonal mean SST biases from 34 CMIP5 models (figure 21). It is clear from this comparison that ECMWF-IFS-LR, which has no operational analog, is compromised by the cold bias in the North Atlantic such that northern hemisphere SST biases are similar to those in the worst performing CMIP5 models. In contrast, zonal mean SST biases in ECMWF-IFS-MR and ECMWF-IFS-HR, which are close to the operational configuration of the ECMWF seasonal forecast model, are comparable to the best-performing CMIP5 models and very similar to the CMIP5 multi-model mean at latitudes north of 40°S.

## 5.4   The value of multi-decadal integrations

The experiments performed for this study are atypical for ECMWF. However, multi-decadal climate runs are an important ingredient for a seamless approach to Earth system model development for weather and climate forecasting. For example, from the comparisons in section 3.3 and additional sensitivity experiments with the ECMWF seasonal forecast system (not shown), it is clear that the impact of ocean resolution is time-scale dependent. This means that biases identified from initialized (re-
)forecasts are not always a reliable indicator of the asymptotic behaviour of the coupled model, and in fact may depend more on the quality of the initial conditions provided by reanalyses. Furthermore, the larger signals in multi-decadal integrations can provide valuable information about the fidelity of simulated processes that is harder to detect in short-range forecasts. In fact, for an equivalent signal-to-noise ratio in the mean bias, multi-decadal climate integrations are potentially much cheaper than ensemble (re-)forecasts.

The asymptotic behaviour of coupled models is also becoming more important to the numerical weather forecasting community with the development of coupled approaches to earth system reanalysis. The climatological attractor of the coupled model is especially important in such reanalyses because of its influence as a background field for periods and/or regions with limited observational constraints.

The ECMWF-IFS configuration presented here will serve as platform for further research and sensitivity experiments as
part of the HighResMIP and PRIMAVERA projects. The scientific and technical developments required for this study will also have a legacy at ECMWF as multi-decadal coupled climate experiments become more important for the development of Earth system models used for weather forecasting and reanalysis.

*Code and data availability.*   The model configurations described here are based on the ECMWF Integrated Forecast System (IFS) and the NEMO/LIM ocean-sea ice model. The IFS source code is available subject to a license agreement with ECMWF. ECMWF member-state
weather services and their approved partners will be granted access. The IFS code without modules for data assimilation is also available for educational and academic purposes as part of the OpenIFS project (https://software.ecmwf.int/wiki/display/OIFS/OpenIFS+Home). The NEMO/LIM source code is available under a CeCILL free software license (https://www.nemo-ocean.eu/). Model output data will be avail-




able through the European Research Council Horizon 2020 PRIMAVERA project (https://www.primavera-h2020.eu/modelling/data-access/).
Further details regarding model configurations and data availability are available from the authors on request.

*Competing interests.* The authors have no competing interests.

*Acknowledgements.* This work was facilitated by funding from the European Research Council Horizon 2020 project PRIMAVERA (641727).
5  ECMWF provided institutional support and access to high-performance computing. We thank Magdalena Balmaseda for insightful comments
on an earlier version of this manuscript and Kristian Mogensen for help with the development of the ECMWF-IFS configuration.



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



**Table 1.** Differences in the NEMO ocean model configurations used in ECMWF-IFS-LR and ECMWF-IFS-MR/-HR.

| Parameter | ECMWF-IFS-LR | ECMWF-IFS-MR/-HR |
|---|---|---|
| Horizontal grid ($n_x \times n_y$) | ORCA1 ($362 \times 292$) | ORCA025 ($1442 \times 1021$) |
| Number of vertical levels (thickness of top level) | 75 (1 m) | 75 (1 m) |
| Horizontal momentum diffusion (rn_ahm_0) | $1 \times 10^4$ m$^2$/s (Laplacian) | $1 \times 10^{11}$ m$^4$/s (bi-Laplacian) |
| Background vertical eddy viscosity (rn_avm0) | $1.2 \times 10^{-4}$ m$^2$/s | $1.0 \times 10^{-4}$ m$^2$/s |
| Background vertical eddy diffusivity (rn_avt0) | $1.2 \times 10^{-5}$ m$^2$/s | $1.0 \times 10^{-5}$ m$^2$/s |
| Gent and Mcwilliams (1990) parameterization | yes | no |
| Iso-neutral tracer diffusivity (rn_aht_0) | 1000.0 m$^2$/s | 300.0 m$^2$/s |
| Eddy-induced velocity coefficient (rn_aeiv_0) | 1000.0 m$^2$/s | n/a |

**Table 2.** Radiative and turbulent energy fluxes at the lower boundary of the atmosphere in observations representative of the early 21st Century (Wild et al., 2013) and model experiments for the period 2000-2014. Fluxes are positive downwards and expressed in W/m$^2$ relative to Earth's surface area. Values in parentheses represent the range of uncertainties given by Wild et al. (2013)

| | Absorbed solar radiation | Net thermal radiation | Latent heat | Sensible heat | Net ($F_{sfc}$) |
|---|---|---|---|---|---|
| Wild et al. (2013) | 161 (154, 166) | -55 (-60, -50) | -85 (-90, -80) | -20 (-25, -15) | 0.6 (0.2,1.0) |
| ECMWF-IFS-LRA | 161.0 | -57.1 | -85.8[a] | -17.7 | 0.4 |
| ECMWF-IFS-HRA | 162.2 | -57.5 | -86.7[a] | -17.6 | 0.4 |
| ECMWF-IFS-LR | 161.2 | -58.0 | -84.0[a] | -18.0 | 1.2 |
| ECMWF-IFS-HR | 163.3 | -58.3 | -86.[a] | -17.7 | 0.9 |

[a] Includes a flux of approximately -0.9 W/m$^2$ associated with the latent heat of fusion in snow fall

Zhang, J. and Rothrock, D.: Modeling global sea ice with a thickness and enthalpy distribution model in generalized curvilinear coordinates, Monthly Weather Review, 131, 845–861, 2003.

Zhang, T. and Sun, D.-Z.: ENSO asymmetry in CMIP5 models, Journal of Climate, 27, 4070–4093, 2014.

**Appendix A**



**Table A1.** Mapping between ECMWF-IFS HighResMIP experiments and internal ECMWF experiment identifiers.

| Model | HighResMIP experiment | Member | ECMWF IDs |
|---|---|---|---|
| ECMWF-IFS-LR | *spinup-1950* | r1i1p1f1 | guz2 (member=0), gv84 (member=0) |
| ECMWF-IFS-LR | *control-1950* | r1i1p1f1 | gv9j (member=0), gvf9 (member=0), gvrb (member=0) |
| ECMWF-IFS-LR | *hist-1950* | r1i1p1f1 | gv9s (member=0), gvhm (member=0) |
| ECMWF-IFS-LR | *hist-1950* | r2i1p1f1 | gv9p (member=0), gvj3 (member=0) |
| ECMWF-IFS-LR | *hist-1950* | r3i1p1f1 | gv9p (member=1), gvj3 (member=1) |
| ECMWF-IFS-LR | *hist-1950* | r4i1p1f1 | gv9p (member=2), gvj3 (member=2) |
| ECMWF-IFS-LR | *hist-1950* | r5i1p1f1 | gv9p (member=3), gvj3 (member=3) |
| ECMWF-IFS-LR | *hist-1950* | r6i1p1f1 | gv9p (member=4), gvj3 (member=4) |
| ECMWF-IFS-LR | *highresSST-present* | r1i1p1f1 | gqnr (member=0), gr8z (member=0) |
| ECMWF-IFS-LR | *highresSST-present* | r2i1p1f1 | gpwf (member=0), gqjc (member=0) |
| ECMWF-IFS-LR | *highresSST-present* | r3i1p1f1 | gpa6 (member=0), gpun (member=0) |
| ECMWF-IFS-LR | *highresSST-present* | r4i1p1f1 | gpa6 (member=1), gpun (member=1) |
| ECMWF-IFS-LR | *highresSST-present* | r5i1p1f1 | gpa6 (member=2), gpun (member=2) |
| ECMWF-IFS-LR | *highresSST-present* | r6i1p1f1 | gpa6 (member=3), gpun (member=3) |
| | | | |
| ECMWF-IFS-MR | *spinup-1950* | r1i1p1f1 | gude (member=0), gv2r (member=0) |
| ECMWF-IFS-MR | *control-1950* | r1i1p1f1 | gv9u (member=0), gvr7 (member=0), gx04 (member=0) |
| ECMWF-IFS-MR | *hist-1950* | r1i1p1f1 | gv9s (member=0), gvhm (member=0) |
| | | | |
| ECMWF-IFS-HR | *spinup-1950* | r1i1p1f1 | gp6b (member=0), gpup (member=0) |
| ECMWF-IFS-HR | *control-1950* | r1i1p1f1 | gq40 (member=0), gqng (member=0), grdm (member=0) |
| ECMWF-IFS-HR | *hist-1950* | r1i1p1f1 | gqnq (member=0), gr8x (member=0) |
| ECMWF-IFS-HR | *hist-1950* | r2i1p1f1 | gq6f (member=0), gqnj (member=0) |
| ECMWF-IFS-HR | *hist-1950* | r3i1p1f1 | gq5p (member=0), gqnl (member=0) |
| ECMWF-IFS-HR | *hist-1950* | r4i1p1f1 | gq5p (member=1), gqnl (member=1) |
| ECMWF-IFS-HR | *highresSST-present* | r1i1p1f1 | gqns (member=0), grdn (member=0) |
| ECMWF-IFS-HR | *highresSST-present* | r2i1p1f1 | gpwi (member=0), gqnc (member=0) |
| ECMWF-IFS-HR | *highresSST-present* | r3i1p1f1 | gpa8 (member=0), gpuo (member=0) |
| ECMWF-IFS-HR | *highresSST-present* | r4i1p1f1 | gpa8 (member=1), gpuo (member=1) |





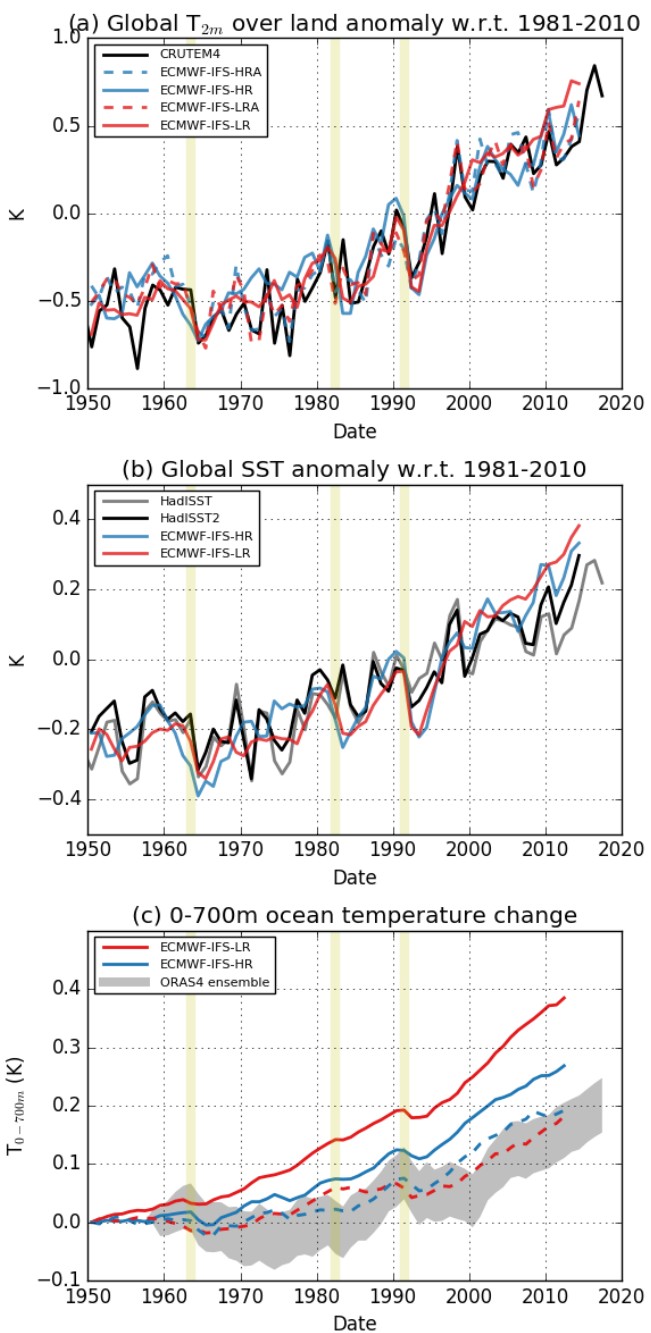

**Figure 1.** (a) Global mean 2 m temperature over land ($T_{2m}$) from ensemble means of the *hist-1950* and *highresSST-present* experiments and the CRUTEM4 observational data set (Jones et al., 2012; Osborn and Jones, 2014). (b) Global mean sea surface temperature (SST) from ensemble means of the *hist-1950* experiment and the HadISST (Rayner et al., 2003) and HadISST2 observational data sets (Rayner et al., 2016; Titchner and Rayner, 2014). (c) Volume-averaged ocean temperature (0-700m) change relative to the first year of data in individual ensemble members of the *hist-1950* experiment and the ORAS4 ensemble of ocean reanalyses (Balmaseda et al., 2013). Dashed lines are estimates of temperature change in *hist-1950* after adjusting for long-term drift in the ocean by calculating temperature change relative to the corresponding *control-1950* experiments. In all plots, periods corresponding to the Mount Pinatubo (1991), El Chichón (1982), and Mount Agung (1963) volcanic eruptions are shaded in yellow. All data points represent annual mean values.



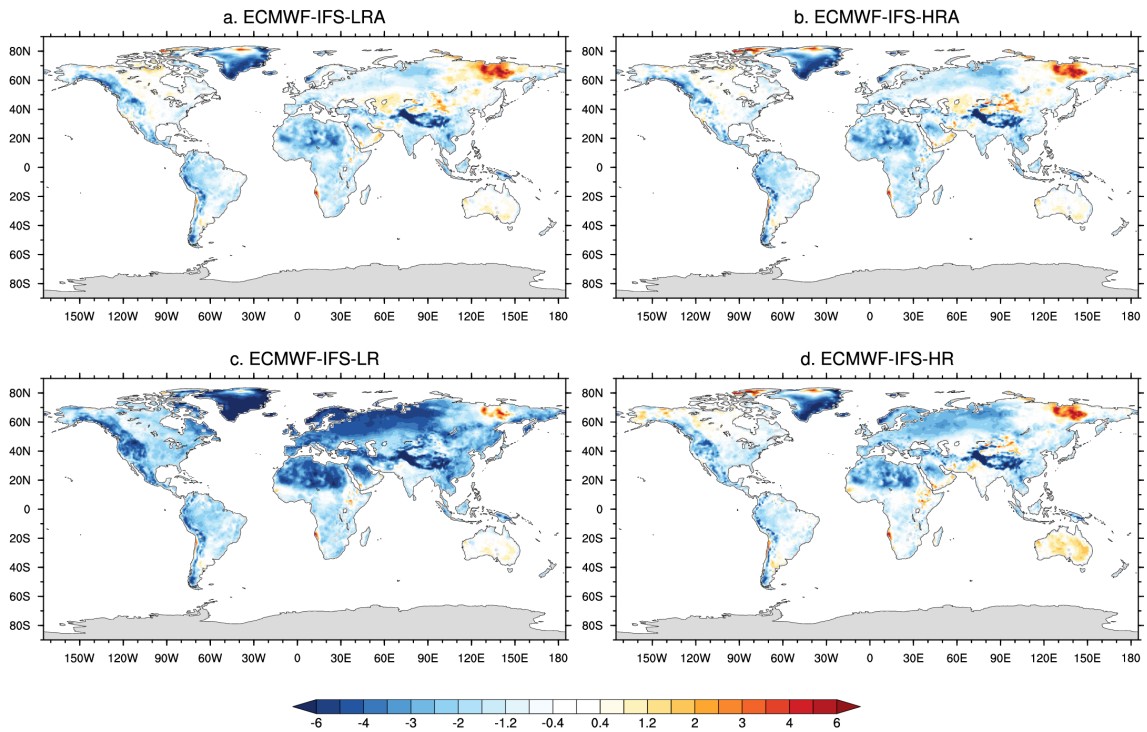

**Figure 2.** Annual mean bias in 2m temperature over land (°C) in (a-b) *highresSST-present* and (c-d) *hist-1950* relative to the CRU TS 4.01
data set (CRU; Harris et al., 2014) for the period 1981-2010.







**Figure 3.** Climatological surface ocean biases calculated using years 1-50 of the *spinup-1950* experiment. (a-c) Sea surface temperature (SST) and (d-f) sea surface salinity (SSS) biases relative to the EN4 climatology representative of the period 1950-1954 that was used to initialize the ocean (Good et al., 2013). (g-i) Annual mean mixed layer depth (MLD) biases relative to the de Boyer Montégut et al. (2004) climatology. (j-l) Net surface heat flux biases (positive downwards) relative to the net heat flux derived from radiative fluxes at the top of the atmosphere combined with the mass-adjusted energy divergence in the ERA-interim reanalysis (Liu et al., 2015, 2017). (m-o) Sea surface height (SSH) biases relative to satellite-derived estimates of absolute dynamic topography (AVISO, 2017).

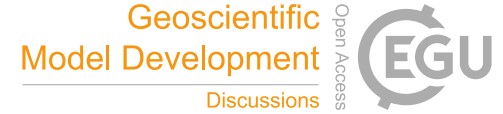

**Mean Bias in Precipitation (mm day$^{-1}$) relative to GPCP 1981-2010**



**Figure 4.** (a-d) Annual mean bias in total precipitation (mm/day) in (a-b) *highresSST-present* and (c-d) *hist-1950* relative to the Global Precipitation Climatology Project data set v2.3 (GPCP; Adler et al., 2003) for the period 1981-2010. (e) Zonal mean of annual mean precipitation (mm/day) in *highresSST-present*, *hist-1950* and the GPCP data set.



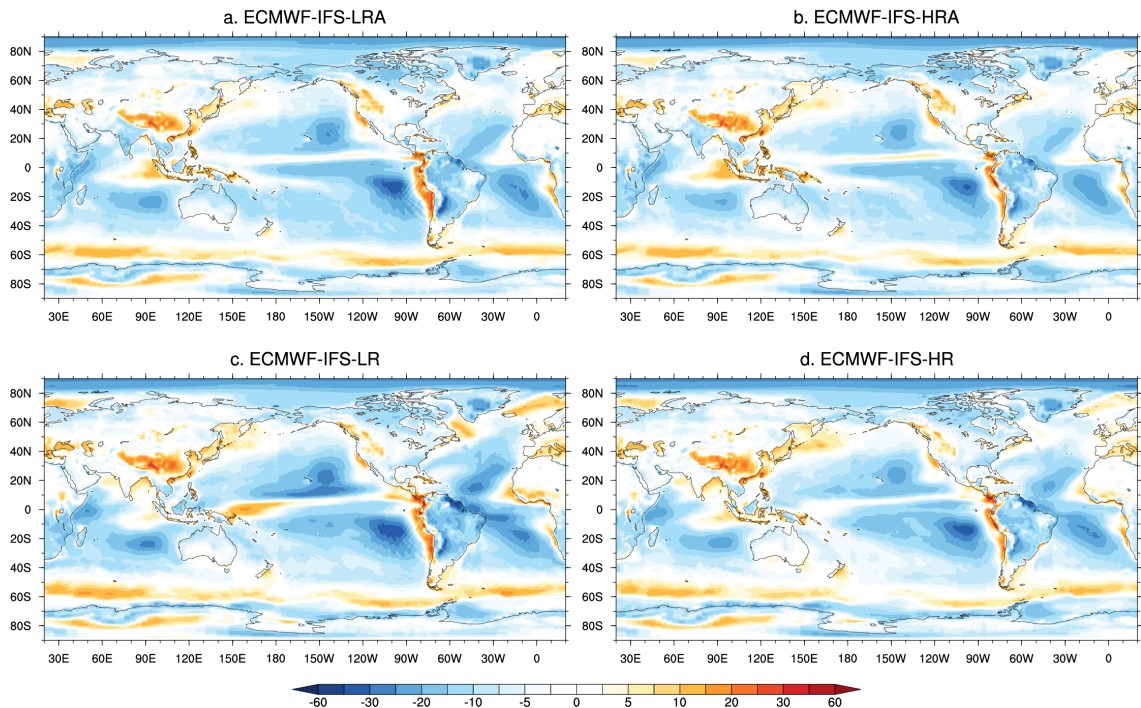

**Figure 5.** Annual mean bias in total cloud radiative forcing (CRF) in (a-b) *highresSST-present* and (c-d) *hist-1950* relative to data from CERES-EBAF Surface Fluxes Edition 4.0 (Kato et al., 2013).



**Mean Bias in SW Cloud Radiative Forcing (W m$^{-2}$) relative to CERES-EBAF 2001-2014**

**Figure 6.** As figure 5 but for short-wave radiation fluxes.





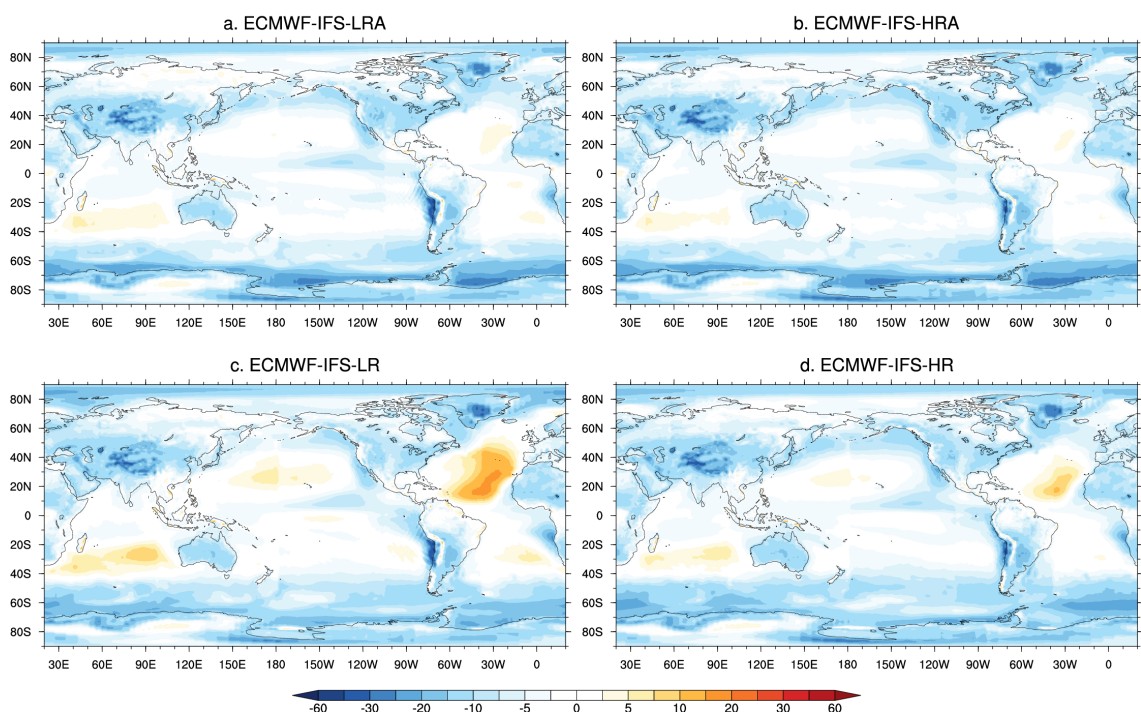

**Figure 7.** As figure 5 but for long-wave radiation fluxes.





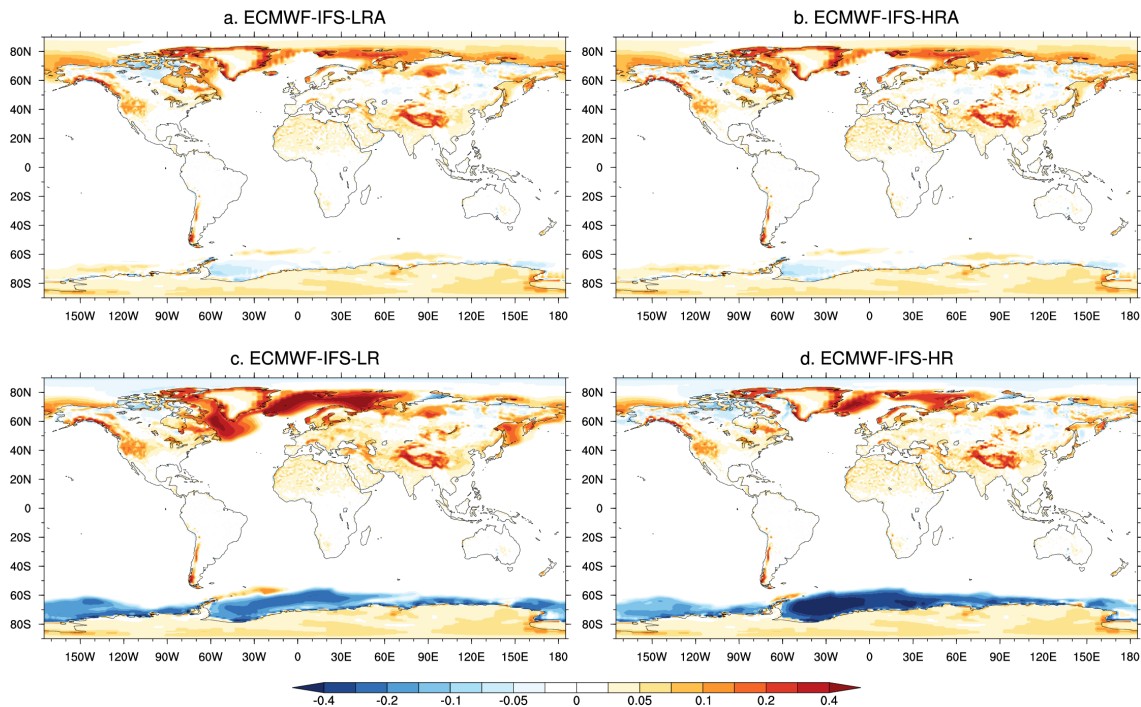

**Figure 8.** Mean bias in short-wave surface albedo in (a-b) *highresSST-present* and (c-d) *hist-1950* relative to estimates derived from CERES-EBAF Surface Fluxes Edition 4.0 (Kato et al., 2013). Surface albedo is defined to be $\alpha_{sfc} = SW_{up}/SW_{down}$, where $SW_{down}$ is the short-wave radiation at the surface and $SW_{up}$ is upward short-wave at the surface.





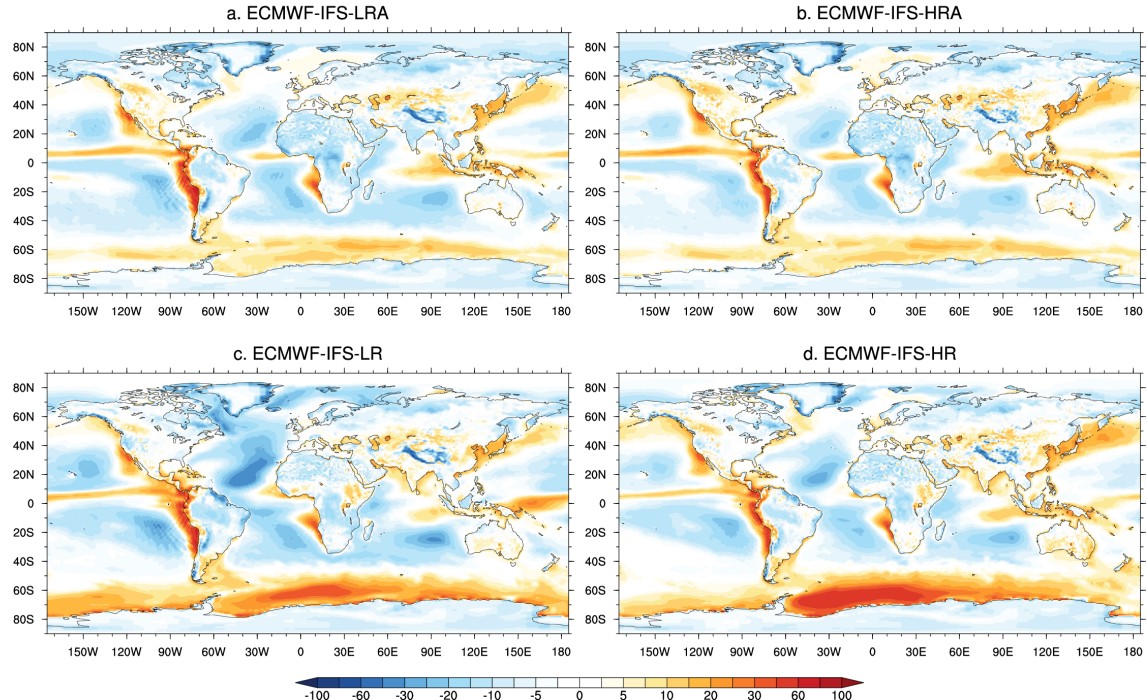

**Figure 9.** Annual mean bias in net surface short-wave radiation in (a-b) *highresSST-present* and (c-d) *hist-1950* relative to data from CERES-EBAF Surface Fluxes Edition 4.0 (Kato et al., 2013).



**Mean-state Bias in Annual Zonal Mean Temperature (°C) relative to ERA-Interim 1981-2010**

**Figure 10.** Mean bias in zonal-mean temperature in (a-b) *highresSST-present* and (c-d) *hist-1950* relative to ERA-Interim for the period 1981-2010 (shaded) and the climatological mean from ERA-Interim (contours).

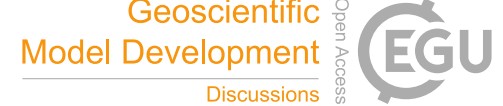



**Figure 11.** Mean bias in zonal-mean wind in (a-b) *highresSST-present* and (c-d) *hist-1950* relative to ERA-Interim for the period 1981-2010 (shaded) and the climatological mean from ERA-Interim (contours).





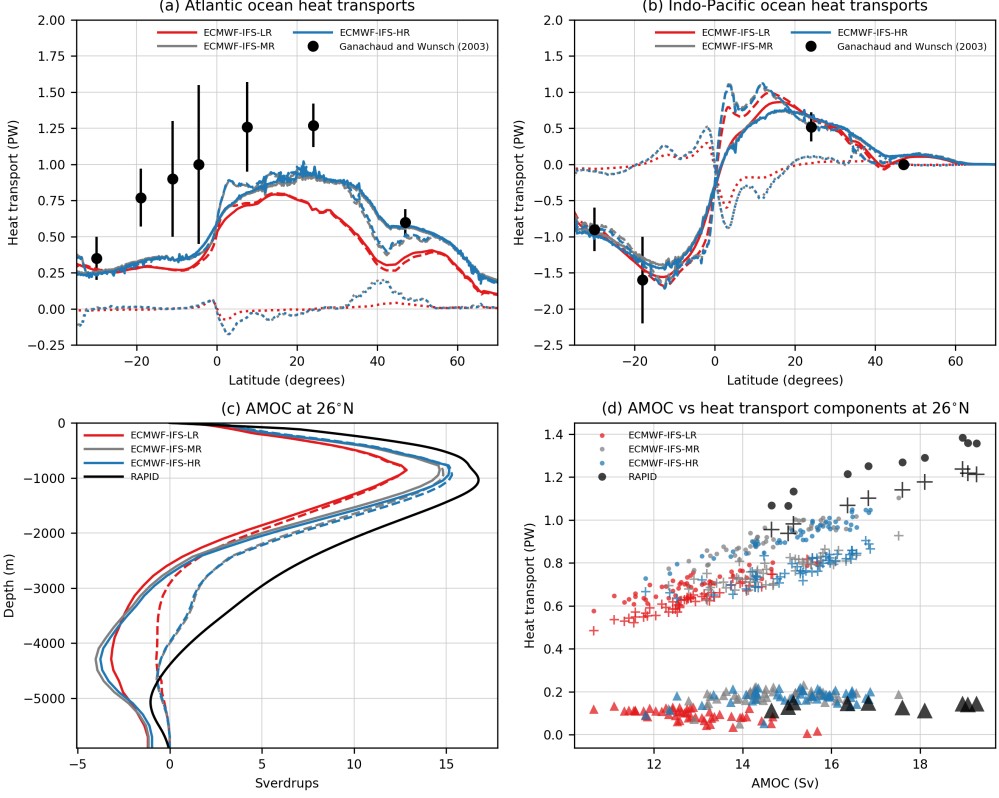

**Figure 12.** (a-b) Meridional ocean heat transports calculated using years 1-50 of the *spinup-1950* experiment compared with the observational estimates from Ganachaud and Wunsch (2003). Total heat transports (solid lines) are decomposed into contributions from the mean flow ($\overline{V} \cdot \overline{T}$; dashed lines) and time-varying 'eddies' ($\overline{V' \cdot T'}$; dotted lines). (c) Stream functions of the Atlantic meridional overturning circulation (AMOC) at 26.5 °N calculated using years 1-50 of *spinup-1950* compared to observations from RAPID-MOCHA array for the period 2004-2015 (McCarthy et al., 2015). Dashed lines correspond to model profiles calculated using the RapidMoc python tool (Roberts, 2017) with a geostrophic reference depth of 4750 m. (d) Strength of the Atlantic meridional overturning circulation (AMOC) at 26.5 °N against total (circles), overturning (plus signs) and horizontal gyre (triangles) components of the meridional heat transport in *spinup-1950*. Observed heat transports are from the RAPID-MOCHA array (Johns et al., 2011) and symbols corresponds to annual means.




**Figure 13.** As figure 3, but calculated for the first year of spinup-1950.





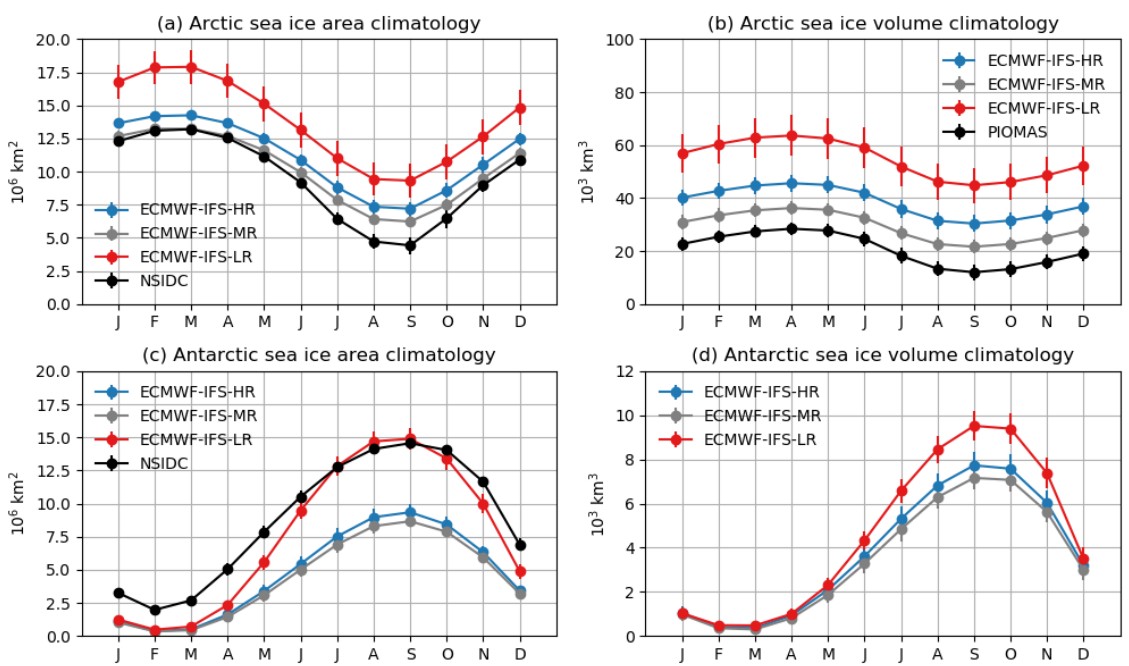

**Figure 14.** Monthly sea-ice area and volume climatologies for the period 1981-2010 from *hist-1950* experiments. Sea ice areas are compared to observational data from the National Snow and Ice Data Centre (NCIDC; Fetterer et al., 2017). Arctic ice volumes are compared with estimates from the Pan-Arctic Ice Ocean Modeling and Assimilation System (PIOMAS; Schweiger et al., 2011; Zhang and Rothrock, 2003).

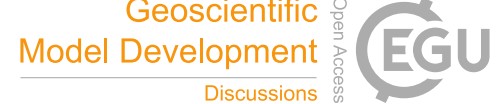



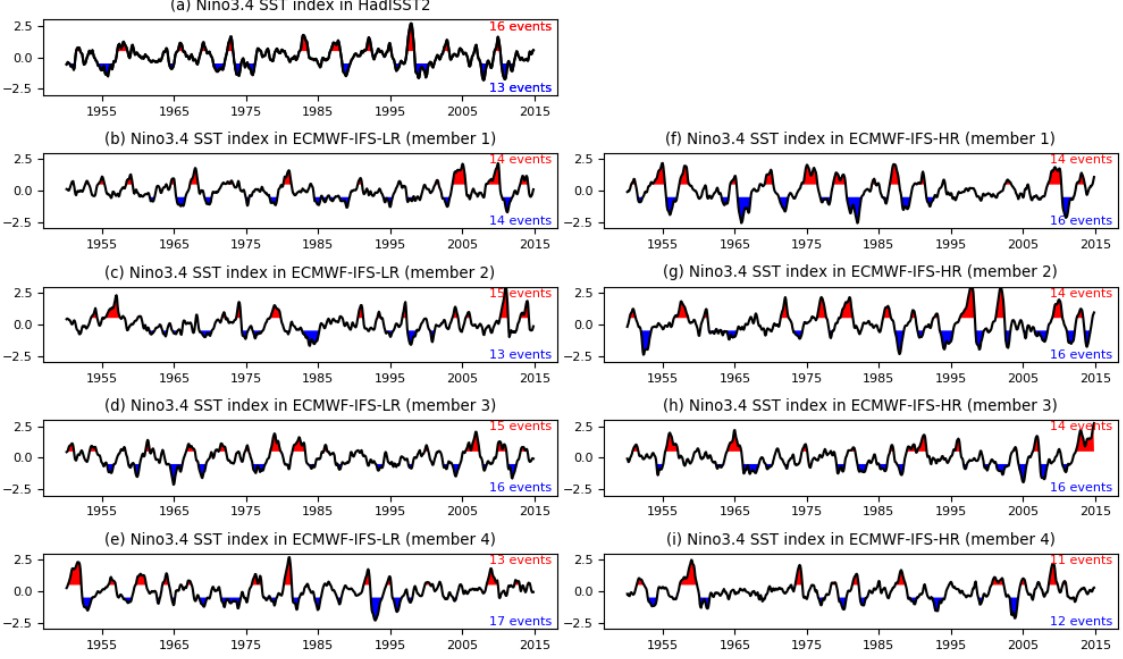

**Figure 15.** Detrended three-month running mean SST anomalies (K) relative to 1981-2010 in the Niño3.4 region (120°W-170°W and 5°S-5°N) from (a) HadISST2, (b-e) four members of *hist-1950* from ECMWF-IFS-LR, and (f-i) four members of *hist-1950* from ECMWF-IFS-HR. Highlighted El Niño/La Niña events correspond to periods with five consecutive and overlapping three-month means with SST anomalies above 0.5 K/below -0.5 K.

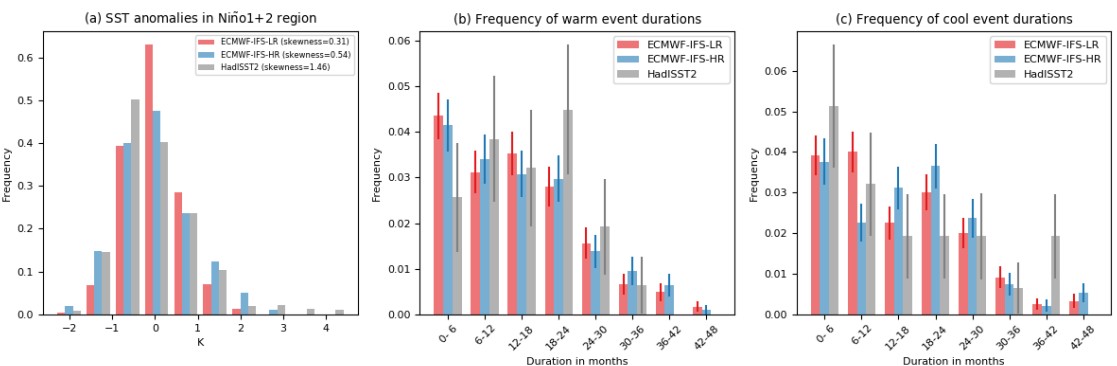

**Figure 16.** (a) Histogram of three-month running mean SST anomalies relative to 1981-2010 in the Niño1+2 region (90°W-80°W and 10°S-0). Frequency of (b) warm and (c) cool event durations determined from the time series plotted in figure 15 and additional ensemble members from *hist-1950*, *control-1950* and *spinup-1950*. Error bars in (b) and (c) represent a bootstrap estimate of the standard error in frequency and are derived by resampling from the available events 10000 times.





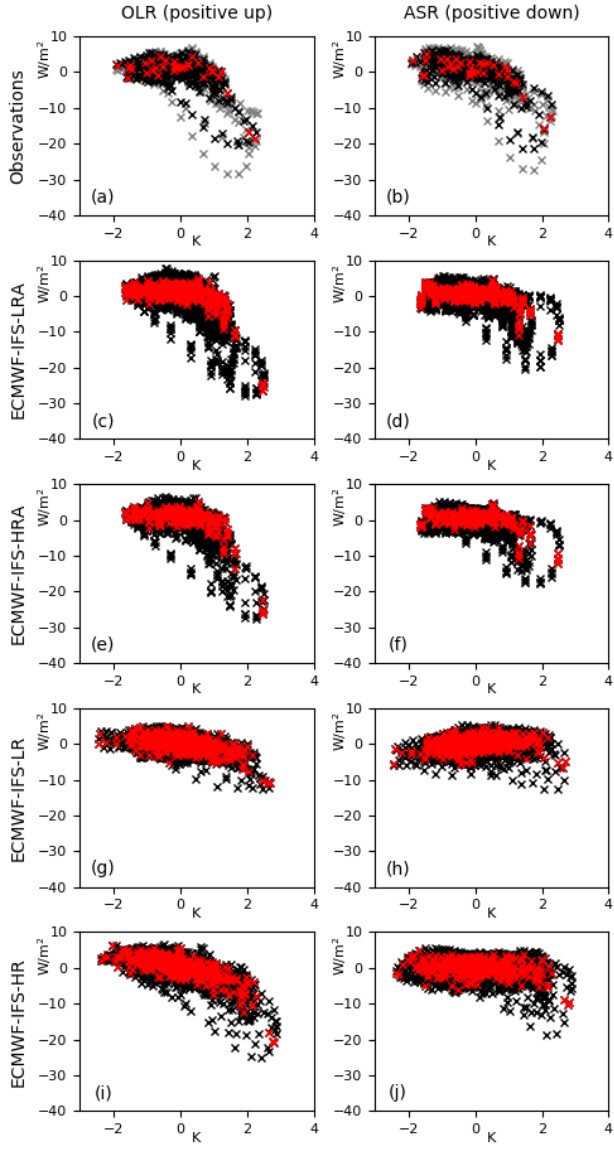

**Figure 17.** Scatter plots of SST anomalies from Niño3.4 region against anomalies of top-of-atmosphere outgoing long-wave radiation (OLR) and top-of-atmosphere absorbed solar radiation (ASR). Observation-based estimates are from a composite of satellite radiation products (black crosses, Mayer et al., 2016) and the ERA-interim reanalysis (grey crosses, Dee et al., 2011). ECMWF-IFS data is for all available HighResMIP experiments and ensemble members. Data are monthly anomalies relative to the 1981-2010 seasonal cycle and all values are smoothed using a 13-point binomial filter prior to plotting. Black crosses represent all monthly values and red crosses are December values, when ENSO-related anomalies are usually strongest.





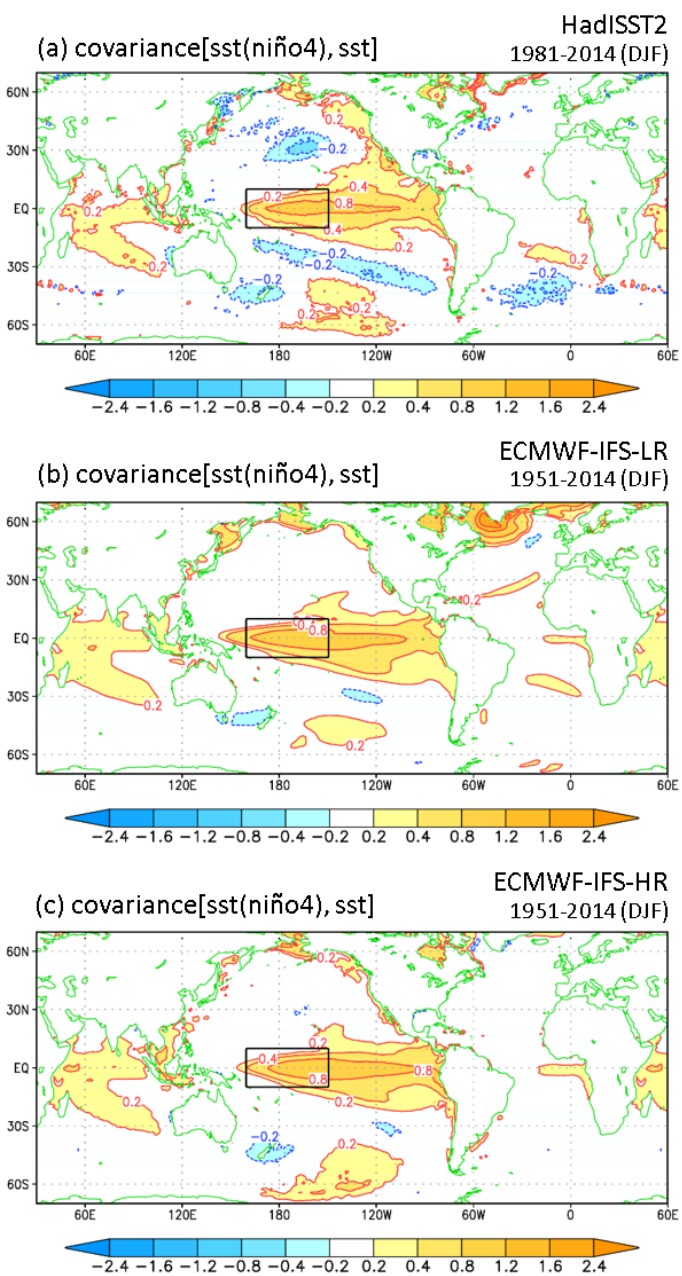

**Figure 18.** Covariance of DJF SST anomalies with the normalized time series of SST anomaly in the Niño4 region for (a) HadISST2 and all members of *hist-1950* in (b) ECMWF-IFS-LR and (c) ECMWF-IFS-HR.





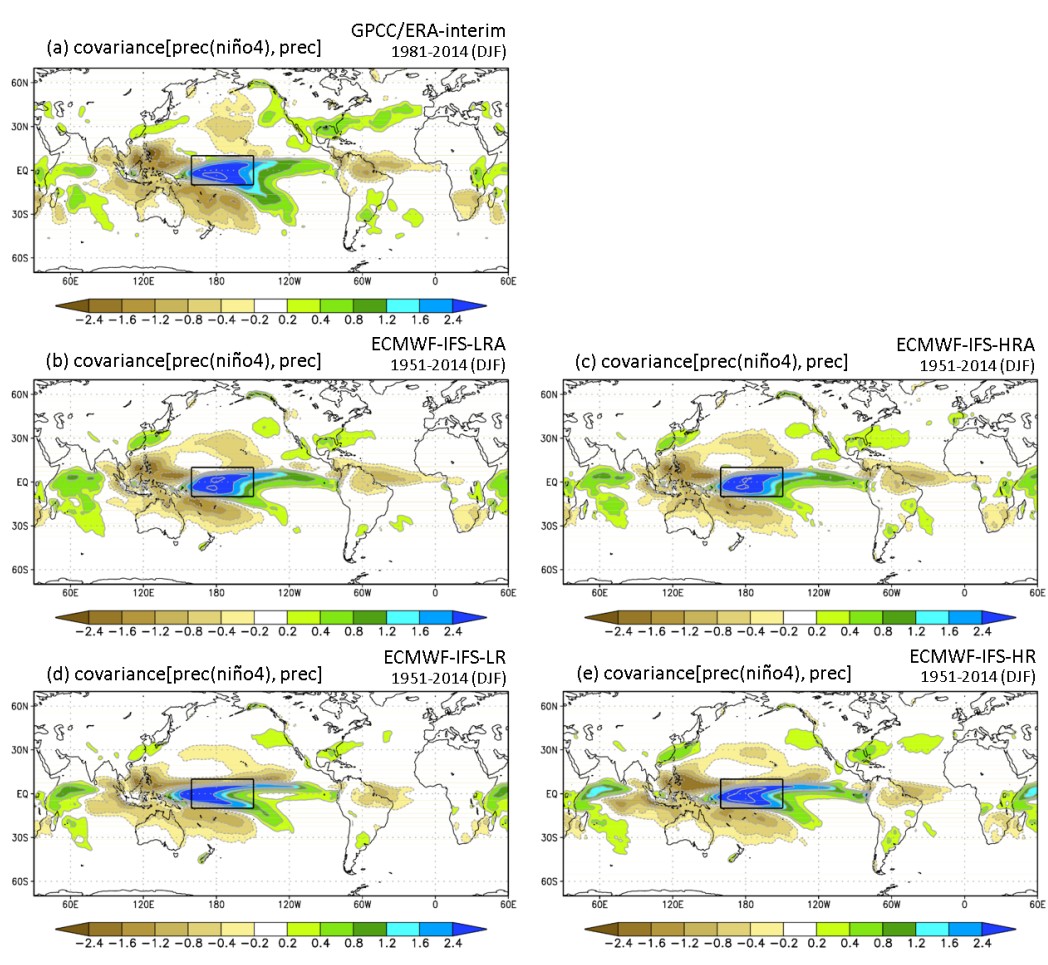

**Figure 19.** Covariance of DJF rainfall anomalies with the normalized time series of rainfall anomaly in the Niño4 region for (a) GPCC observations (Niño4 precipitation) and ERA-interim (global precipitation), all members of *highresSST-present* from (b) ECMWF-IFS-LRA and (c) ECMWF-IFS-HRA, and all members of *hist-1950* from (d) ECMWF-IFS-LR and (e) ECMWF-IFS-HR.





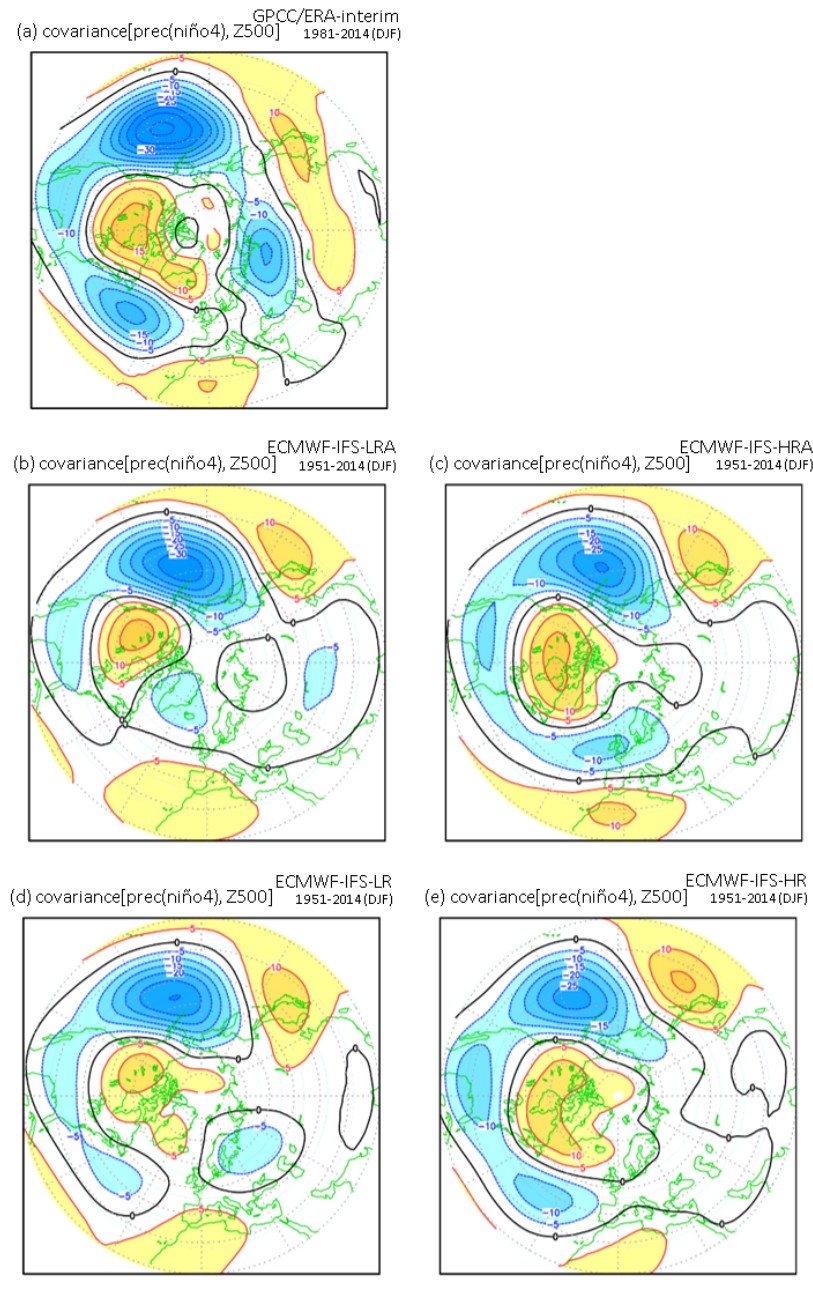

**Figure 20.** As figure 19, but for DJF 500 hPa geopotential height anomalies.

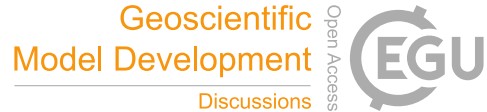



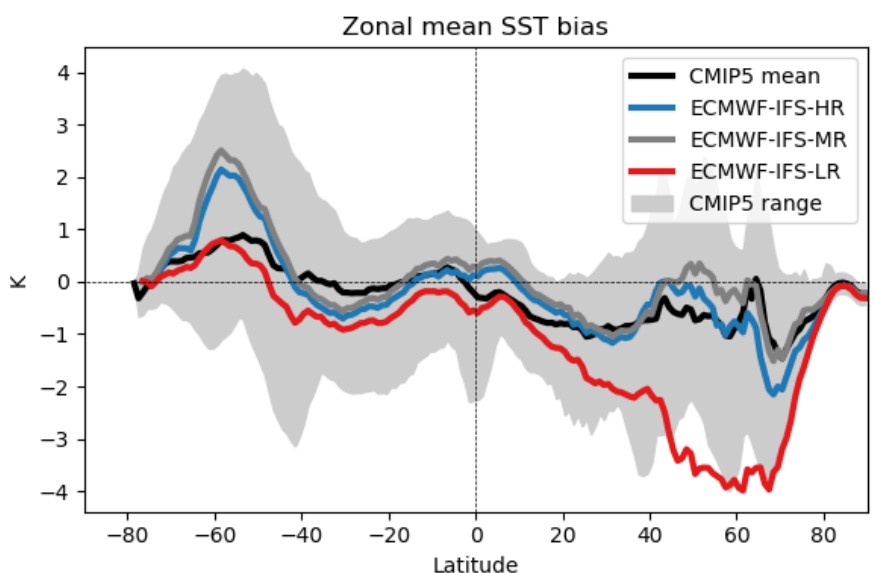

**Figure 21.** Zonal mean SST biases relative to EN4 (Good et al., 2013) for the period 1986-2005 from ECMWF-IFS (experiment *hist-1950*) and CMIP5 models (experiment *historical*).