# Peer review of "Climate model configurations of the ECMWF Integrated Forecast System (ECMWF-IFS cycle 43r1) for HighResMIP"

_Geoscientific Model Development, 2018_

## Referee Comment (RC1) · Anonymous Referee #1 · 28 May 2018

The authors provide a detailed description of the ECMWF-IFS model for climate simulations in HighResMIP. Following the presentation of the different model components and the coupling between them, a comprehensive and honest analysis of the model performance is given, taking into account a wide range of aspects from regional to global scale.

The scientific quality of the model description and analysis are excellent, and require no changes except for a few minor technicalities. Given the nature of this contribution, the scientific innovation is limited. An important aspect highlighted in this work, often ignored in the scientific community, is that model calibrations based on "short" periods

of one year +/- are insufficient for multi-decadal simulations.

I recommend publication after considering the following minor modifications:

(1) Page 3, lines 29-30: the time-steps for the two model configurations are large (36s/km for Tco199, 48s/km for Tco399) compared to other modeling systems. It may be worth mentioning that this is due to the different nature of the IFS model (semi-implicit, spectral) than models that the scientific community is likely more familiar with (e.g. WRF, split-explicit, typically 6s/km time-step). Also, if experiments were made with smaller or larger time-steps, it would be nice to mention any differences here. Related to this: page 5, section coupling: if experiments were made with different coupling frequencies, it would be nice to briefly mention this here.

(2) Page 4, ines 11-15: I would recommend adding one sentence on how the 1km landuse classification is mapped onto the model grid and how dominant vegetation categories are derived.

(3) Page 7, lines 13-14: One important aspect of contributions to (not only, but in particular) this journal is the possibility for readers/reviewers to reproduce the results. I believe that readers trying to do so will need more information on the seed differences (see also comment (8) below).

(4) Page 7, sections 2.8.1 and 2.8.2: Although these are partly mentioned later in the text, I would recommend adding a statement with typical spin-up times (vs the spin-up times used here) to each of the sections (2.8.1: atmosphere and land/soil properties; 2.8.2 ocean and ice).

(5) Page 11, lines 27-28: Is it possible to describe briefly what the "dynamic origin" for the temperature errors could be? It seems to me from looking at the figures that the most significant temperature biases referred to in this last section are correlated with coastlines close to significant orography (e.g. the Andes).

(6) Page 12, lines 30-32: Are the deficiency in the TKE scheme and the statement that

an increase in vertical resolution leads to better results (lines 4-5, same page) related? If so, it would be nice to make this connection here.

(7) Page 15, line 21: I believe that there are two commas missing: "shorter, more intense" and "longer, less intense".

(8) Page 19, section "Code and data availability": As mentioned before, I believe that readers should have the possibility to reproduce the results (assuming they have access to the software). In addition to the present description of where to obtain the model code, I would recommend making the changes to the model configuration available (either as instructions in the abstract or to download, or as a patch set to download).

General comments:

(1) Related to the last comment on section "Code and data availability", I am missing a short statement on the computational requirements of running ECMWF-IFS. Where were the current simulations made, what were the required resources? What would be minimal and typical resources needed to run the model (#nodes or cores, main memory, temporary/permanent disk space)? I am not asking for a detailed model performance analysis, just a few numbers that help potential users to estimate the resources that they may need for their experiments. Given the large time-step of the model, I believe that the computational resources are lower than for other coupled modeling systems, which makes the ECMWF-IFS an attractive alternative.

(2) The number of figures is close to overwhelming. I believe that some/few (e.g. figure 13) could be left out and I invite the authors the consider this for their final version.

---

## Referee Comment (RC2) · Anonymous Referee #2 · 5 Jun 2018

Review of "Climate model configurations of the ECMWF Integrated Forecast System (ECMWF-IFS cycle 43r1) for HighResMIP" by Roberts, Senan, Molteni, Boussetta, Mayer and Keeley, submitted to Geosci. Model Dev.

This paper looks at the climate modelling skill of the ECMWF Integrated Forecast System. (ECMWF-IFS). ECMWF-IFS is typically only used for forecasts up to a year ahead. There is an interest to see if it can also perform climate-length simulations. The experiments follow the HighResMIP protocol and are based on the PRIMAVERA project.

The paper is very thorough and realistic about the skill of the ECMWF-IFS and its

sensitivity to resolution. A 1deg. ocean and 0.25deg. ocean are considered, and the atmosphere is at either 50km or 25km. As well as the high and low resolution configurations, a mixed resolution run with low-res atmosphere and high-res ocean is employed. I abbreviate the runs to LR, HR and MR.

This is the first paper that I am aware of that arose from HighResMIP, and I think it is a very valuable contribution to the literature, being honest about improvements and degradations due to resolution. This is useful for the community in the choosing between e.g. larger ensembles or higher resolution, to get a more reliable indication of the climate system characteristics.

Comments General comment: I think some reference to earlier papers discussing sensitivity of climate models to resolution should be made, both in the Introduction, and their findings may be useful for the main results section. See e.g. (but not limited to) McClean et al 2011, Delworth et al. 2012, Bacmeister et al. 2014, Small et al. 2014, Griffies et al. 2015.

Page 3, Line 29. I think this should be ECMWF-IFS-HR (uses Tco399 grid)

Page 5, Line 16. It would be useful to know why the run-off from land is not coupled. Is this not important for short-term forecasts i.e. the main use of the model?

Page 9, lines 5-10. The results for the adjusted 0-700m ocean heat content change are indeed quite impressive (after removing the drift in CTL). Was such a good fit (to ORAS4) expected? This positive result should be mentioned in Conclusions and Abstract.

Figs 6, 9. It is interesting that the coupled model biases in SST in eastern boundary regions do not seem to largely affect the CRF or net surface shortwave. (Compare uncoupled and coupled results)

Page 12, line 20. You could add here the warm SST bias at eastern boundaries is common to all resolutions but with slightly differing magnitude (see Small et al. 2015

for some discussion of sensitivity to resolution.)

Page 12, lines 25-30. In Fig. 12a,b, is the eddy part for LR got from the GM parameterization? (Please clarify in text.) Is the eddy part for MR shown in Figs 12a-b ? Finally, in the text please point out the latitudes of the improvements in the eddy heat transport. For the Atlantic it seems that eddy transport may contribute to differences between models at around 40deg. N, but does not contribute much between 10deg.N and 30deg.N. So presumably the mean flow is quite different between models in the latter latitude range.

Fig. 3,13. In the N. Atlantic, LR may have a slightly larger SST bias in year 1 (Fig. 13), which may be amplified under coupled feedbacks in the long term (Fig. 3) whereas for MR and HR the coupled feedbacks may be less important ? You might want to add an additional figure showing close-up of N. Atlantic for the SST bias in year 1 and long-term.

Also for Fig. 3, 13, I was a bit surprised about the lack of sensitivity of MLD to ocean resolution, following the work of Lee et al. 2011. Firstly you might want to use an updated MLD climatology such as Holte et al 2017, Roemmich et al 2009 using the vastly better sampling of ARGO. Please show a close up of the S. Ocean, showing MLD for LR-ARGO, MR-ARGO and HR-ARGO, all for the key winter season, July-August-September.

I'm not sure if this is mentioned in the text, but it is very notable from Figs 3,13 that the biases in SST, SSS, SSH in the S. Ocean are a slow process (but MLD biases appear immediately). Any thoughts on this?

It might be worth mentioning that MR and HR have excessive deep mixed layers in the Labrador Sea (but again see comment on using a more recent observed product.)

Regarding the very weak ACC in MR and HR (half as much as observed) is this simply due to the reduced temperature gradient between the poles and subtropics (from the

[Figure]

warm SST bias at high latitudes), and a consequence of geostrophy/thermal wind, or is it something more complicated?

Fig. 16b, c – it looks like the model has some events longer than any observed (e.g. 42-48 months). Perhaps this is just due to the short observed record, such that longer events may be possible but not recently observed.

Page 18, lines 12-13 and then lines 19-21 are worded very similarly so that I was confused as to whether this was a sentence erroneously repeated, or a real result. Can they be reworded ?

Page 19, lines 16-19. I was confused by these sentences. Can they be clarified and explained in more detail?

References

Bacmeister, J. T., M. F. Wehner, R. B. Neale, A. Gettelman, C. Hannay, P. H. Lauritzen, J. M. Caron and J. E. Truesdale (2014), Exploratory High-Resolution Climate Simulations using the Community Atmosphere Model (CAM), J. Clim., 27, 3073–3099.

Delworth, T. L., A. Rosati, W. Anderson, A. J. Adcroft, V. Balaji, R. Benson, K. Dixon, S. M. Griffies, H.-C. lee, R. C. Pacanowski, G. A. Vecchi, A. T. Wittenburg, F. Zeng, and R. Zhang (2012), Simulated climate change in the GFDL CM2.5 high-resolution coupled climate model, J. Clim., 25, 2755-2781.

Griffies, S. and 12 co-authors (2015), Impacts on ocean heat from transient mesoscale eddies in a hierarchy of climate models, J. Clim.

Holte, J., Talley, L. D., Gilson, J., & Roemmich, D. (2017). An Argo mixed layer climatology and database. Geophysical Research Letters, 44(11), 5618–5626. https://doi.org/10.1002/2017GL073426

Lee, M.-M., AJ. G. Nurser, I. Stevens, and J.-B. Sallée, 2011. Subduction over the Southern Indian Ocean in a high-resolution atmosphere-ocean coupled model. J.

Clim., 24, 3830-3849.

McClean, J., D. C. Bader, F. O. Bryan, P. W. Jones, J. Dennis, A. Mirin, M. Vertenstein, D. P. Ivanova, M. E. Maltrud, Y.-Y. Kim, J. Boyle, N. Norton, A. Craig, R. Jacob and P. Worley (2011), A prototype two-decade fully-coupled fine-resolution CCSM simulation, Ocean Modeling, 39, 10-30.

Roemmich, D., & Gilson, J. (2009). The 2004–2008 mean and annual cycle of temperature, salinity, and steric height in the global ocean from the Argo Program. Progress in Oceanography, 82(2), 81–100. https://doi.org/10.1016/j.pocean.2009.03.004

Small, R. J., and 18 co-authors, 2014. A new synoptic scale resolving global climate simulation using the Community Earth System Model, J. Adv. Model. Earth Syst., 06, doi:10.1002/2014MS000363.

Small, R. J., E. Curchitser, K. Hedstrom, B. Kauffman, and W. G. Large 2015. The Benguela upwelling system: quantifying the sensitivity to resolution and coastal wind representation in a global climate model. J. Climate, 28, 9409-9432. http://journals.ametsoc.org/doi/abs/10.1175/JCLI-D-15-0192.1

---

## Author Comment (AC1) · 31 Jul 2018

We thank the referees for their useful comments. We have made almost all of the recommended changes, including modifications to the text and updating one of the observational data sets used to produce figures 3 and 13. We have attached a pdf document containing a tracked-changes version of our manuscript and point-by-point response to each comment.

Regards, Chris Roberts (on behalf of all authors)

Please also note the supplement to this comment:

[Figure]

https://www.geosci-model-dev-discuss.net/gmd-2018-90/gmd-2018-90-AC1-supplement.pdf

---

## Author Response (AR1)

**Response to reviews for "Climate model configurations of the ECMWF Integrated Forecast System (ECMWF-IFS cycle 43r1) for HighResMIP"**

We thank the referees for their useful comments. We have made almost all of the recommended changes, including modifications to the text and updating one of the observational data sets used to produce figures 3 and 13. We have attached a tracked-changes version of our manuscript and our point-by-point response to each comment is included below.

**Referee #1**

**(1) Page 3, lines 29-30: the time-steps for the two model configurations are large (36s/km for Tco199, 48s/km for Tco399) compared to other modeling systems. It may be worth mentioning that this is due to the different nature of the IFS model (semi-implicit, spectral) than models that the scientific community is likely more familiar with (e.g. WRF, split-explicit, typically 6s/km time-step). Also, if experiments were made with smaller or larger time-steps, it would be nice to mention any differences here. Related to this: page 5, section coupling: if experiments were made with different coupling frequencies, it would be nice to briefly mention this here.**
We have made the recommended changes and have added the following sentences to section 2.1:

*"The comparatively long time-steps used in the IFS are enabled by the unconditionally stable semi-Lagrangian advection coupled with semi-implicit time-stepping. Reducing the time-step from 1200 s to 900 s in the Tco399 configuration had a minimal impact on model biases and forecast skill on seasonal timescales (not shown)."*

The coupling frequency is the same in all configurations and is specified in section 2.5.
*"fluxes are exchanged between sub-models with a coupling frequency of 1 hour in all configurations"*

**(2) Page 4, ines 11-15: I would recommend adding one sentence on how the 1km land use classification is mapped onto the model grid and how dominant vegetation categories are derived.**
We have added the following sentence to section 2.2:

*"For each model grid box, the fractional coverage and vegetation model parameters are then set according to the dominant vegetation type within the GLCC data set."*

**(3) Page 7, lines 13-14: One important aspect of contributions to (not only, but in particular) this journal is the possibility for readers/reviewers to reproduce the results. I believe that readers trying to do so will need more information on the seed differences (see also comment (8) below).**
We have made the recommended change and added the following reference that describes the SPPT scheme in detail:

*"Ensemble members are distinguished by different seeds to the SPPT scheme, where the seed determines the initial random two-dimensional fields that are used to perturb the parameterized model tendencies (Leutbecher et al., 2017)."*

**(4) Page 7, sections 2.8.1 and 2.8.2: Although these are partly mentioned later in the text, I would recommend adding a statement with typical spin-up times (vs the spin-up times used here) to each of the sections (2.8.1: atmosphere and land/soil properties; 2.8.2 ocean and ice).**
We have made the recommended change and added the following text to sections 2.8.1 and 2.8.2:

*"There is no separate spin-up integration for the highresSST-present ensemble. This approach is acceptable for the majority of atmosphere and land-surface variables that adjust quickly (< 1 year) to the imposed forcings. One notable exception is the water content of the deepest soil layers (72-199 cm), which takes several years to reach a new equilibrium."*

*"The 50 year spin-up period follows the HighResMIP protocol and is sufficient for the near-surface ocean and sea-ice to reach an approximate steady state. In contrast, the ocean interior takes many centuries to fully adjust and is still drifting at the end of the 50 year spinup integration."*

**(5) Page 11, lines 27-28: Is it possible to describe briefly what the "dynamic origin" for the temperature errors could be? It seems to me from looking at the figures that the most significant temperature biases referred to in this last section are correlated with coastlines close to significant orography (e.g. the Andes).**

As recommended by the reviewer we have added a sentence to note the potential importance of changes in orography.

"In other locations, near-surface cold biases are associated with positive biases in net surface short-wave radiation suggesting a dynamic origin for such temperature errors. This effect is most pronounced for the Andean mountains, which could be indicative of errors in the response of the atmospheric circulation to steep orography."

**(6) Page 12, lines 30-32: Are the deficiency in the TKE scheme and the statement that an increase in vertical resolution leads to better results (lines 4-5, same page) related?**
**If so, it would be nice to make this connection here.**

These issues are not related as they concern vertical resolution in the atmosphere and vertical mixing in the ocean. We have clarified this in the text by explicitly specifying "vertical resolution in the atmosphere" and "TKE scheme for vertical mixing in the ocean".

**(7) Page 15, line 21: I believe that there are two commas missing: "shorter, more intense" and "longer, less intense".**
We have made the suggested changes.

**(8) Page 19, section "Code and data availability": As mentioned before, I believe that readers should have the possibility to reproduce the results (assuming they have access to the software). In addition to the present description of where to obtain the model code, I would recommend making the changes to the model configuration available (either as instructions in the abstract or to download, or as a patch set to download).**

We completely support the sentiment that all such experiments should be reproducible by the scientific community. Unfortunately, the IFS scripts and source code cannot be made publicly available as a supplementary material due to licensing constraints. In addition, the operational nature of the suites used to perform these experiments means that it is non-trivial to provide a simple configuration file that can be used to repeat our experiments elsewhere. However, we are happy to provide the required information to reproduce results on the ECMWF systems on a case-by-case basis and have modified our code availability statement to reflect this:

*"Further details regarding data availability and model configurations, including the information required to reproduce our results on ECMWF systems, are available from the authors on request."*

**(1) Related to the last comment on section "Code and data availability", I am missing a short statement on the computational requirements of running ECMWF-IFS. Where were the current simulations made, what were the required resources? What would be minimal and typical resources needed to run the model (#nodes or cores, main memory, temporary/permanent disk space)? I am not asking for a detailed model performance analysis, just a few numbers that help potential users to estimate the resources that they may need for their experiments. Given the large time-step of the model, I believe that the computational resources are lower than for other coupled modeling systems, which makes the ECMWF-IFS an attractive alternative.**
We have made the recommended change and added a table summarizing the computational cost of coupled model configurations to the appendix.

**(2) The number of figures is close to overwhelming. I believe that some/few (e.g. figure 13) could be left out and I invite the authors the consider this for their final version.**
Given the nature of this work and the scope of the GMD journal we would prefer to keep the figures in the main text rather than move some to a supplementary document.

**Referee #2**

**General comment: I think some reference to earlier papers discussing sensitivity of climate models to resolution should be made, both in the Introduction, and their findings may be useful for the main results section. See e.g. (but not limited to) McClean et al 2011, Delworth et al. 2012, Bacmeister et al. 2014, Small et al. 2014, Griffies et al. 2015.**
We have made the suggested changes and added the following additional text to the relevant locations in the introduction and results sections.

*"Previous studies have shown that increases in ocean and atmospheric model resolution can affect many aspects of the climate system, including ocean and atmospheric biases (Roberts et al., 2009; Gent et al., 2010; Small et al., 2014), ENSO variability (Shaffrey et al., 2009; Delworth et al., 2012), ocean fronts and western boundary currents (Kirtman et al., 2012; Chassignet and Xu, 2017), the representation of tropical cyclones (McClean et al., 2011), the nature of air-sea interaction at the ocean mesoscale (Bryan et al., 2010), and the global ocean heat budget (Griffies et al., 2015)."*

*"This is consistent with the results of McClean et al. (2011), who found that increasing atmospheric resolution in the Community Climate System Model had little impact on the large scale radiation biases."*

*"The limited response to a change in atmospheric resolution is not unique to the ECMWF model. For example, Bacmeister et al. (2014) found that the mean climate of the Community Atmosphere Model did not dramatically improve when increasing atmospheric model resolution from ~100 km to ~25 km."*

**Page 3, Line 29. I think this should be ECMWF-IFS-HR (uses Tco399 grid)**
Thank you for spotting this error – it has been corrected.

**Page 5, Line 16. It would be useful to know why the run-off from land is not coupled. Is this not important for short-term forecasts i.e. the main use of the model?**
We have modified the sentence to clarify this point:

*"One coupled interaction that is not represented, due to its limited importance at operational time-scales, is the link between precipitation over land and runoff into the ocean."*

**Page 9, lines 5-10. The results for the adjusted 0-700m ocean heat content change are indeed quite impressive (after removing the drift in CTL). Was such a good fit (to ORAS4) expected? This positive result should be mentioned in Conclusions and Abstract.**
We have made the recommended changes and added the following text to the conclusions and abstract:

*"All configurations successfully reproduce the observed long-term trends in global mean surface temperature. Furthermore, following an adjustment to account for 'drift' in the sub-surface ocean, coupled configurations of ECMWF-IFS realistically reproduce observation-based estimates of ocean heat content change since 1950. "*

*"Although developed primarily for weather forecasting timescales, we have shown that ECMWF-IFS also has utility for multi-decadal climate applications. In particular, we have shown that coupled configurations of ECMWF-IFS can successfully reproduce the observed long-term trends in global mean surface temperature and, following an adjustment to account for 'drift' in the sub-surface ocean, realistically reproduce observation-based estimates of ocean heat content change since 1950."*

**Figs 6, 9. It is interesting that the coupled model biases in SST in eastern boundary regions do not seem to largely affect the CRF or net surface shortwave. (Compare uncoupled and coupled results). Page 12, line 20. You could add here the warm SST bias at eastern boundaries is common to all resolutions but with slightly differing magnitude (see Small et al. 2015 some discussion of sensitivity to resolution.)**
We have made the recommended changes and added the following text to section 3.3:
*"In addition, all coupled configurations exhibit positive SST biases along the western coastlines of the South American and African continents associated with a positive bias in short-wave CRF that originates in the atmospheric model (figure 6). The magnitude of these biases shows some sensitivity to ocean model resolution, which could be related to the representation of coastal upwelling that is known to be important for the development of SST biases in such regions (Small et al., 2015)."*

**Page 12, lines 25-30. In Fig. 12a,b, is the eddy part for LR got from the GM parameterization? (Please clarify in text.) Is the eddy part for MR shown in Figs 12a-b ? Finally, in the text please point out the latitudes of the improvements in the eddy heat transport. For the Atlantic it seems that eddy transport may contribute to differences between models at around 40deg. N, but does not contribute much between 10deg.N and 30deg.N. So presumably the mean flow is quite different between models in the latter latitude range.**

We have made the recommended changes added the following text to section 3.3:

*"One of the most important contributors to these improvements is the better representation of meridional heat transports in the higher resolution ocean model. Figures 12a-b show total meridional heat transports decomposed into contributions from the mean flow (i.e. $\overline{V} \cdot \overline{T}$) and transient 'eddies' (i.e. $\overline{V' \cdot T'}$), where overbars indicate a time-mean and primes indicate deviations from a time-mean. Note that the parameterized eddy-induced velocity is included in V for ECMWF-IFS-LR. From these plots it is evident that increasing ocean model resolution can have a large impact on heat transports associated with both the mean flow and the transient eddies."*

**Fig. 3,13. In the N. Atlantic, LR may have a slightly larger SST bias in year 1 (Fig. 13), which may be amplified under coupled feedbacks in the long term (Fig. 3) whereas for MR and HR the coupled feedbacks may be less important ? You might want to add an additional figure showing close-up of N. Atlantic for the SST bias in year 1 and long-term.**
Given the comments from the reviewer #1 we do not want to add any additional figures at this stage.

**Also for Fig. 3, 13, I was a bit surprised about the lack of sensitivity of MLD to ocean resolution, following the work of Lee et al. 2011. Firstly you might want to use an updated MLD climatology such as Holte et al 2017, Roemmich et al 2009 using the vastly better sampling of ARGO. Please show a close up of the S. Ocean, showingMLD for LR-ARGO, MR-ARGO and HR-ARGO, all for the key winter season, July-August-September.**
As stated in the reply above, we prefer not to add any additional figures to the manuscript. However, we have replotted figures 3 and 13 using data from the Holte et al. 2017 climatology. The estimated biases in the Northern Hemisphere are very similar but there are some changes in the Southern Ocean (in fact our model biases are reduced) and we have updated to the text to reflect this.

**I'm not sure if this is mentioned in the text, but it is very notable from Figs 3,13 that the biases in SST, SSS, SSH in the S. Ocean are a slow process (but MLD biases appear immediately). Any thoughts on this?**
We have added an additional sentence noting the differing responses in MLD and SST/SSH:
*"For instance, the main features of the mixed layer depth biases in the Southern Ocean are established within the first year of integration highlighting the dominant role of 'fast' dynamical processes. This rapid response can be contrasted with the more slowly developing SST and SSH biases in the same location, which take longer to develop because they represent a thermodynamic response with timescales governed by the heat capacity of the upper ocean."*

**It might be worth mentioning that MR and HR have excessive deep mixed layers in the Labrador Sea (but again see comment on using a more recent observed product.)**
As suggested, we have updated the observational estimates of MLD and added the following text noting the different biases in the Labrador Sea:

*"...all models exhibit an overly shallow AMOC (figure 12c). This common bias in the depth-structure of the AMOC occurs in spite of contrasting biases in the depth of convection in the Labrador Sea (compare figures 3g-i) and is likely a consequence of issues in the representation of the density-driven Nordic Sea overflows that are known to affect z-coordinate ocean models (Danabasoglu et al., 2010)."*

**Regarding the very weak ACC in MR and HR (half as much as observed) is this simply due to the reduced temperature gradient between the poles and subtropics warm SST bias at high latitudes), and a consequence of geostrophy/thermal wind, or is it something more complicated?**
The reduced ACC is not directly linked the geostrophic adjustment to the reduced gradients in temperature and SSH. We have added the following text to clarify this:

*"This reduced transport through the Drake Passage in ECMWF-IFS-MR/-HR is not a simple geostrophic response to the reduced latitudinal gradients in SSH and ocean temperature. It is instead related to the development of a westward flow of ~50 Sv that occurs between between 64◦ S and 60◦ S, which counters the expected eastward transport of ~120 Sv that occurs between 60◦ S and 56◦ S."*

**Fig. 16b, c – it looks like the model has some events longer than any observed (e.g. 42-48 months). Perhaps this is just due to the short observed record, such that longer events may be possible but not recently observed.**
We have noted this discrepancy and added the following sentence:
*"We also see a small number of simulated events that last longer than any observed event (> 42 months), which could be an artefact of the limited length of the observational record."*

**Page 18, lines 12-13 and then lines 19-21 are worded very similarly so that I was confused as to whether this was a sentence erroneously repeated, or a real result. Can they be reworded ?**
They are separately highlighting the impact of coupling and atmospheric resolution, but they very similarly worded. We have reworded the second sentence to read as follows:

"Increased atmospheric resolution also seems to improve teleconnections between tropical Pacific rainfall and geopotential height anomalies in the North Atlantic, but further work is required to assess the robustness of this result (figure 20)."

**Page 19, lines 16-19. I was confused by these sentences. Can they be clarified and explained in more detail?**
We have rephrased these sentences to read as follows:

[revised manuscript text omitted]